**Subject Category:**
Biology (whole organism)

cellular biology/evolution

three-dimensional reconstruction, F-actin, myosin, functional morphology, lobopod, Ecdysozoa

**Author for correspondence:**
Vladimir Gross
e-mail: vladimir.gross@uni-kassel.de

# Cellular morphology of leg musculature in the water bear *Hypsibius exemplaris* (Tardigrada) unravels serial homologies

## Vladimir Gross and Georg Mayer

Department of Zoology, Institute of Biology, University of Kassel, Heinrich-Plett-Straße 40, 34132 Kassel, Germany

VG, 0000-0001-7422-9148; GM, 0000-0003-0737-2440

Tardigrades (water bears) are microscopic, segmented ecdysozoans with four pairs of legs. Lobopodous limbs that are similar to those seen in tardigrades are hypothesized to represent the ancestral state of Panarthropoda (Tardigrada + Onychophora + Arthropoda), and their evolutionary history is important to our understanding of ecdysozoan evolution. Equally important is our understanding of the functional morphology of these legs, which requires knowledge of their musculature. Tardigrade musculature is well documented but open questions remain. For example, while the muscular organization of each trunk segment and its legs is unique, three of the four trunk segments are nevertheless relatively homonomous. To what extent, then, do leg muscles show segmental patterns? Specifically, which leg muscles are serially repeated and which are unique? The present study addresses these questions using a combination of techniques intended to visualize both the overall layout and fine structure of leg muscles in the eutardigrade *Hypsibius exemplaris*. In doing so, we propose serial homologies for all leg muscles in each of the four legs and reveal new details of their cellular structure and attachment sites. We compare our results to those of previous studies and address the functional implications of specialized muscle cell morphologies.

## 1. Introduction

Tardigrades, commonly known as water bears, comprise a monophyletic group of microscopic ecdysozoans. While they are closely related to arthropods (moulting animals with jointed appendages), their external morphology most closely resembles

that of the extant onychophorans (velvet worms) as well as several extinct lobopodians known from the fossil record [1,2]. Tardigrades rely on short, unjointed, typically clawed appendages called lobopods for locomotion, which are remarkably similar in appearance to those of onychophorans despite differences in body size and habitat—onychophorans are exclusively terrestrial while tardigrades are aquatic or semiterrestrial. This resemblance, together with the hypothesis that lobopods represent the ancestral locomotory appendages of Panarthropoda [3,4], fuels an incentive to study the functional morphology and, by extension, the musculature of the legs.

The muscular system of tardigrades has been well documented since the first detailed studies of tardigrade anatomy over a century ago [5–15]. Through careful examination of histological and whole-mount preparations, early researchers described the fine structure of tardigrade musculature in astonishing detail, even noting that a single muscle cell is composed of different regions [5–8]. Remarkably, many of these details were later confirmed when the ultrastructure of tardigrade muscles and their attachment sites was clarified using transmission electron microscopy [16–19]. The most recent investigations of myoanatomy have relied mainly on fluorescent labelling of filamentous actin—a major component of muscle fibres—in combination with confocal laser-scanning microscopy (CLSM) to generate high-resolution, three-dimensional datasets [11–15]. This in turn has allowed researchers to map the muscles that make up this organ system and compare them between tardigrade species. In doing so, clear patterns common to all tardigrade species have emerged. In all tardigrades examined to date, for example, the somatic musculature can be subdivided into dorsal, lateral and ventral muscle groups (summarized in [20]). In the ventral group, muscles invariably attach at seven points arranged medially along the anterior–posterior body axis: two sites each in the segments bearing legs II–IV and one in the segment bearing leg I [15]. The dorsal and lateral longitudinal muscle strands span the length of the body from the head to the posteriormost body segment, while dorsoventral muscles bridge the dorsal and ventral attachment sites [14,20].

Nonetheless, there is some disagreement between the myoanatomical descriptions of tardigrades in the literature, even between closely related species. For instance, although tardigrades are clearly segmented animals, there is uncertainty in the exact degree of metamerism (i.e. serial repetition along the anterior–posterior body axis) of their body musculature [11,14,15,21]. Segmentation hypotheses in many myoanatomical studies are based largely on the longitudinal and dorsoventral somatic musculature and the morphology of their attachment points [10,14] with some support from the legs and their muscles, which are externally reflective of segmentation [21]. However, the low total number of body segments and the modified posteriormost leg pair complicates the assessment of serial homologies, and consequently, the extent of segmentation of tardigrade musculature ranges in the literature from none or little [11,22] to true of all muscle groups [14].

Another aspect that varies between descriptions is the musculature of the posteriormost leg pair. While all studies agree that the posteriormost leg is associated with the fewest muscles, the total number is debated [11,15]. Whether these and other discrepancies are attributable to species differences or perhaps result from different techniques is unclear. In our view, this discussion may therefore benefit from a combined examination of both the organization of the leg musculature as well as the fine structure of individual muscle cells, which to date has been done only for the marine eutardigrade *Halobiotus crispae* [12].

In order to understand the variation between different myoanatomical descriptions and to address the question of serial homologies, we reinvestigated the leg musculature of the model eutardigrade *Hypsibius exemplaris*. Using markers against both actin and myosin combined with super-resolution CLSM and three-dimensional reconstruction, we map every muscle in each of the four legs and assess the serial homologies thereof. We also performed ultrastructural analysis in order to examine the fine structure of individual muscle cells and their attachment sites. We compare our results to those of previous investigations of the same and related tardigrade species and discuss the functional implications of specialized muscle cell morphologies.

## 2. Material and methods

Animals were maintained as described by Gross *et al.* [23]. Briefly, tardigrades were kept in plastic Petri dishes in mineral water (Volvic, Danone Waters Deutschland GmbH, Frankfurt am Main, Germany) at 21°C and fed unicellular algae (*Chlorococcum* sp.). In preparation for each experiment, specimens of *H. exemplaris* Gąsiorek *et al.* [24] were isolated from algae by rinsing with tap water through a 50 μm nylon mesh and anaesthetized by asphyxiation in tap water at 60°C for 30 min.

## 2.1. Immunohistochemistry and DNA and F-actin labelling

The musculature of adult tardigrades was analysed using a combination of markers. For all experiments involving fluorescent markers, anaesthetized adults were immersed in 4% formalin buffered with PBS-Tx (0.1 mol l$^{-1}$ phosphate-buffered saline, pH 7.4, plus 1% Triton X-100) and fixed for 30 min–4 h at room temperature. During the fixation period, the cuticle of each specimen was punctured using electrolytically sharpened tungsten needles.

For fluorescent labelling of myosin, the samples were incubated in 50 mmol l$^{-1}$ NH$_4$Cl for 30 min following fixation in order to quench fixation-induced autofluorescence. Following several washes in PBS-Tx (2 × 10 min, 2 × 30 min, 2 × 1 h), the animals were incubated in a blocking solution containing 10% NGS (normal goat serum; Sigma-Aldrich, St Louis, MO, USA) in PBS-Tx for 1 h at room temperature. The samples were then incubated in a solution containing mouse anti-myosin II monoclonal antibody (1 : 10 dilution) in PBS-Tx + 1% NGS + 0.02% NaN$_3$ overnight at room temperature. The monoclonal antibody (mouse anti-myosin II, DSHB Product 56-396-5), deposited by Günther Gerisch [25], was obtained from the Developmental Studies Hybridoma Bank (product 56-396-5), created by the NICHD of the NIH and maintained at The University of Iowa, Department of Biology, Iowa City, IA 52242. The specimens were then washed several times throughout the day in PBS-Tx, then incubated in a solution containing goat anti-mouse polyclonal antibody coupled to Alexa Fluor® 568 (1 : 500 dilution; Thermo Fischer Scientific, Waltham, MA, USA) in PBS-Tx + 1% NGS + 0.02% NaN$_3$ overnight at room temperature. Following several washes in PBS-Tx, the specimens were incubated in PBS containing 1 ng ml$^{-1}$ DAPI (4′,6-diamidino-2-phenylindole; Thermo Fischer Scientific) for 15 min, rinsed quickly in PBS and mounted between two glass coverslips in ProLong Gold (Invitrogen, Carlsbad, CA, USA). The coverslips were sealed the following day using nail polish. Negative controls, whereby the primary antibody is omitted from the staining protocol, result in an absence of staining.

For fluorescent labelling of filamentous actin (F-actin), fixed tardigrades were rinsed several times in PBS-Tx and incubated in a solution containing phalloidin-rhodamine (25 µg ml$^{-1}$ in PBS; Thermo Fischer Scientific) overnight at 4°C. On the following day, the samples were rinsed with PBS and mounted as described above. All fluorescently labelled specimens were imaged using either a Leica TCS STED (Leica Microsystems, Wetzlar, Germany) or a Zeiss LSM880 Airyscan (Carl Zeiss GmbH, Jena, Germany) confocal laser-scanning microscope.

## 2.2. Western blots

In order to confirm the specificity of the myosin II antibody, western blots were performed according to the following protocol modified from Hering *et al*. [26] (electronic supplementary material, figure S1). Several Petri dish cultures of *H. exemplaris* were concentrated by washing through a 50 µm nylon mesh and collected in a 1.5 ml centrifuge tube. After adding up to 500 µl of 5× Laemmli buffer and a protease inhibitor (cOmplete™, EDTA-free; Sigma-Aldrich), the tube was placed on ice and the tardigrades were manually homogenized using a plastic homogenizer. The contents were then heated for 10 min at 95°C, sonicated at 24 kHz using a UP200S Ultrasonic Processor (Hielscher Ultrasonics GmbH, Teltow, Germany) and heated again for 10 min. Proteins were separated using SDS-PAGE on a 7.5% gel for 45 min at 200 V. The proteins were transferred to a Porablot NCP nitrocellulose membrane (Macherey-Nagel GmbH & Co. KG, Düren, Germany) via wet blot and blocked with a solution containing 4% powdered milk in PBS for 30 min at room temperature. The samples were then incubated overnight with mouse anti-myosin II antibody (1 : 5 dilution), washed 6 × 5 min with PBS, incubated with goat anti-mouse antibody conjugated with alkaline phosphatase (1 : 10 000 dilution; dianova GmbH, Hamburg, Germany) and washed again with PBS. The signal was developed using a solution containing 50 µg ml$^{-1}$ BCIP (5-bromo-4-chloro-3-indolyl phosphate; Thermo Fischer Scientific) in dimethylformamide. The reaction was stopped by transferring to distilled water. In order to judge blotting efficiency, the original gel was subsequently stained with Brilliant Blue G 250 (Carl Roth GmbH, Karlsruhe, Germany).

## 2.3. Electron microscopy

For scanning electron microscopy (SEM), anaesthetized specimens were fixed in 4% formalin buffered with PBS overnight at 4°C. After several rinses in PBS, the specimens were transferred to a capsule with a pore size of 78 µm and dehydrated through an ascending ethanol series (10 min each at 30%,

50%, 70%, 90%, 2 × absolute ethanol). The capsule containing tardigrades was then transferred in absolute ethanol to a BAL-TEC CPD 030 critical point dryer (Balzers, Liechtenstein) and dried. Individual tardigrades were transferred to double-sided carbon tape on aluminium stubs using an eyebrow hair. In order to expose the inside of the tardigrade, one stub containing several specimens was pressed against a second, empty stub with carbon tape and the two were pulled apart, splitting the specimens. All samples were then sputter coated with approximately 30 nm of gold-palladium using a Polaron SC7640 sputter coater (Quorum Technologies, Kent, UK) and analysed using a Hitachi S-4000 field emission scanning electron microscope (Hitachi High-Technologies Europe GmbH, Krefeld, Germany) at an accelerating voltage of 5 kV.

Ultrastructural analysis was performed via scanning transmission electron microscopy (STEM). For this purpose, asphyxiated specimens were fixed in 4% formalin + 1% glutaraldehyde in 0.067 mol l$^{-1}$ Sørensen phosphate buffer [27] for 1 h at room temperature. After 2 × 15 min washes in buffer, the samples were post-fixed in 1% $OsO_4$ in Sørensen buffer for 1 h at room temperature. The samples were then washed several times quickly and then overnight in distilled water. On the following day, the samples were dehydrated through an increasing ethanol series as described above followed by 2 × 10 min in acetone. The tardigrades were then embedded in epoxy resin using a Spurr Low-Viscosity Embedding Kit (Sigma-Aldrich) as follows: Specimens were first incubated in a 1 : 1 mixture of acetone and Spurr resin (using the manufacturer's 'standard' formulation) for 30 min at room temperature, followed by pure Spurr resin for 2 × 1 h and then overnight. On the following day, the specimens were transferred to fresh Spurr resin and poured into plastic box-shaped moulds. The specimens were oriented individually within the moulds using fine needles and polymerized at 70°C for 3 days. Sectioning was done with glass knives using a Reichert-Jung Ultracut E ultramicrotome (C. Reichert AG, Vienna, Austria). Serial sections displaying silver–gold interference colours were collected on single-slot copper grids coated with Formvar®/Vinylec® (Plano GmbH, Wetzlar, Germany) and contrasted for 5 min with a uranyl acetate substitute (UAR-EMS, Electron Microscopy Sciences, Hatfield, PA, USA; 1 : 4 dilution in 50% methanol [28]) and for 8 min with lead citrate [29]. Samples were imaged on a custom-made STEM-in-SEM grid holder using a Hitachi S-4000 field-emission scanning electron microscope at an acceleration voltage of 30 kV. Images of serial sections were aligned using the TrakEM2 plugin for FIJI [30–32].

## 2.4. Image processing and segmentation

Raw Airyscan image stacks were processed in Zen Blue (Carl Zeiss) with filter strength set to 'auto'. CLSM substacks were assembled and adjusted for brightness and contrast using the FIJI distribution of ImageJ2 [33]. The SEM images were adjusted for brightness and contrast and artificially coloured using Adobe Photoshop CS6 (Adobe Systems, San Jose, CA, USA). Specimens that showed an even, high-quality fluorescent signal were subsequently chosen for segmentation, which was done manually using Amira 5.4.0 (Thermo Fischer Scientific). For creating three-dimensional representations, surfaces of the individually segmented muscles were superimposed over a volume-rendering of the original CLSM stack. For creating CLSM substacks featuring individual artificially coloured muscles, the exported label fields of segmented muscles were used as a mask over the original CLSM data. All three-dimensional images and CLSM substacks shown are of only one side of the body. Electronic supplementary material, movies were created using Amira 5.4.0 and assembled using Adobe Premiere Pro CS6 (Adobe Systems). All diagrams, labelling and assembly of image plates were done using Adobe Illustrator CS6 (Adobe Systems).

# 3. Results

## 3.1. Nomenclature and morphology of muscles and their attachment sites

Fluorescent labelling of F-actin revealed the entire body musculature of *H. exemplaris*. Our analysis is focused on the leg musculature, which accounts for a large proportion of the total body musculature (figures 1*a,b*, 2, 3, 4*a–e* and 5*a–d*; electronic supplementary material, movies S1 and S2). Combined fluorescent labelling of myosin II, F-actin and DNA show that some muscles contain two or more contractile strands but only a single nucleus (figures 6*a–d* and 7*a–d*). The nucleus belonging to each muscle is identifiable via myosin labelling, which stains both the contractile strand and the cell body surrounding the nucleus (figures 6*a–d* and 7*a–d*). In these cases where a muscle cell branches, each

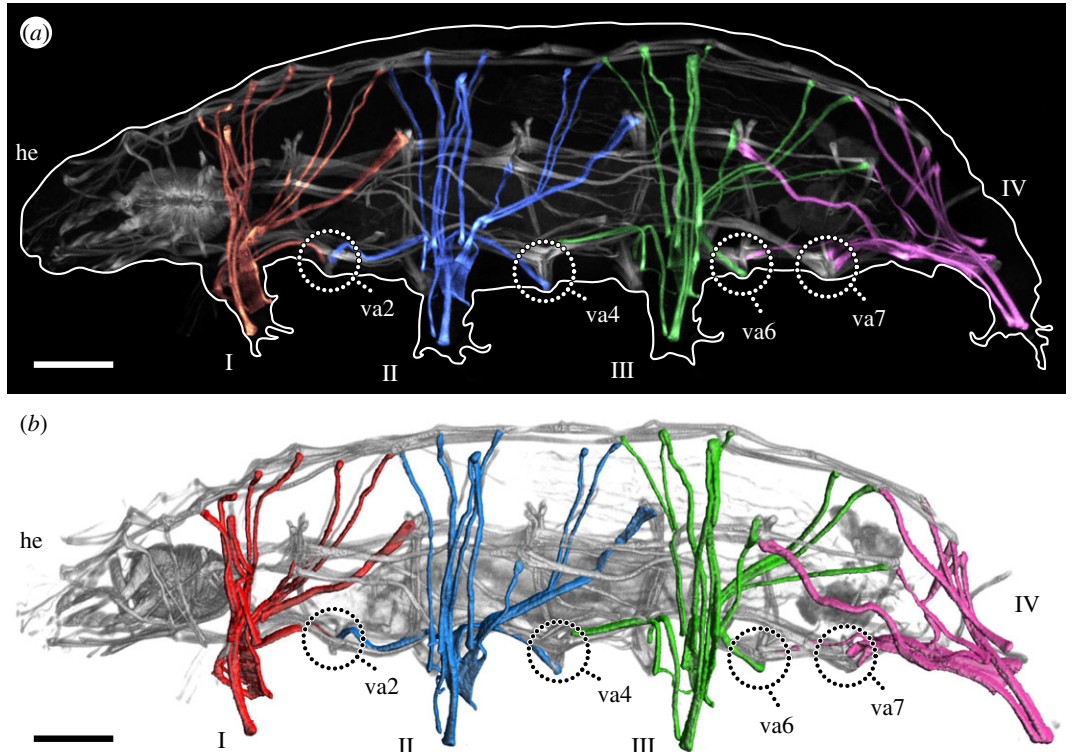

**Figure 1.** Overview of the muscular system of the eutardigrade *H. exemplaris*. Colour-coding according to leg. Lateral view; anterior is left, dorsal is up in both images. Note that ventral attachment sites 1, 3 and 5 are located at the same level as legs I, II and III, respectively, and are therefore not labelled. (*a*) CLSM substack showing the muscular system of the left half of the animal, with leg muscles artificially coloured by leg. F-actin labelling. (*b*) Three-dimensional reconstruction based on the CLSM dataset shown in (*a*). he, head; I–IV, legs I–IV; va2–va7, ventral attachment sites 2–7. Scale bars, 20 µm.

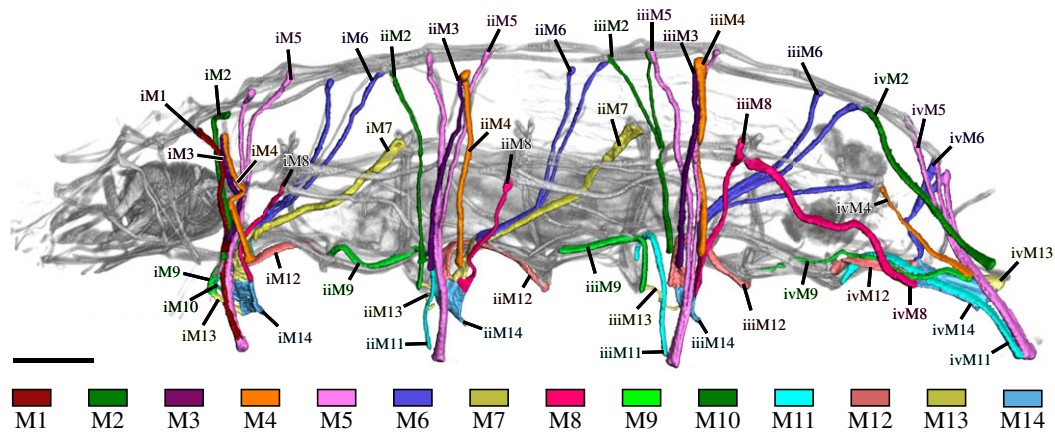

**Figure 2.** Leg muscles of the tardigrade *H. exemplaris*. Colour-coding according to hypothesized serial homologues. Three-dimensional reconstruction based on CLSM data of F-actin labelling. Lateral view; anterior is left, dorsal is up. Scale bar, 20 µm.

strand has its own proximal attachment site, but all strands merge distally. This construction is confirmed by SEM of bisected specimens (figure 6*a*). In this work, we therefore define an individual muscle based on the presence of one nucleus and consider several strands that share a nucleus as a single muscle.

Since a single muscle may have several attachment sites and a single attachment site may be shared by several muscles, we created a nomenclature that is independent of attachment sites in order to avoid ambiguity (see electronic supplementary material, table S1 for a comparison with previous nomenclatures). In naming leg muscles, we use specific labels for hypothesized serially homologous muscles (table 1). Each muscle is designated by the leg it supplies (i, ii, iii or iv) followed by its specific number (e.g. iM2 for muscle M2 of the first leg, iiM2 for muscle M2 of the second leg, etc.)

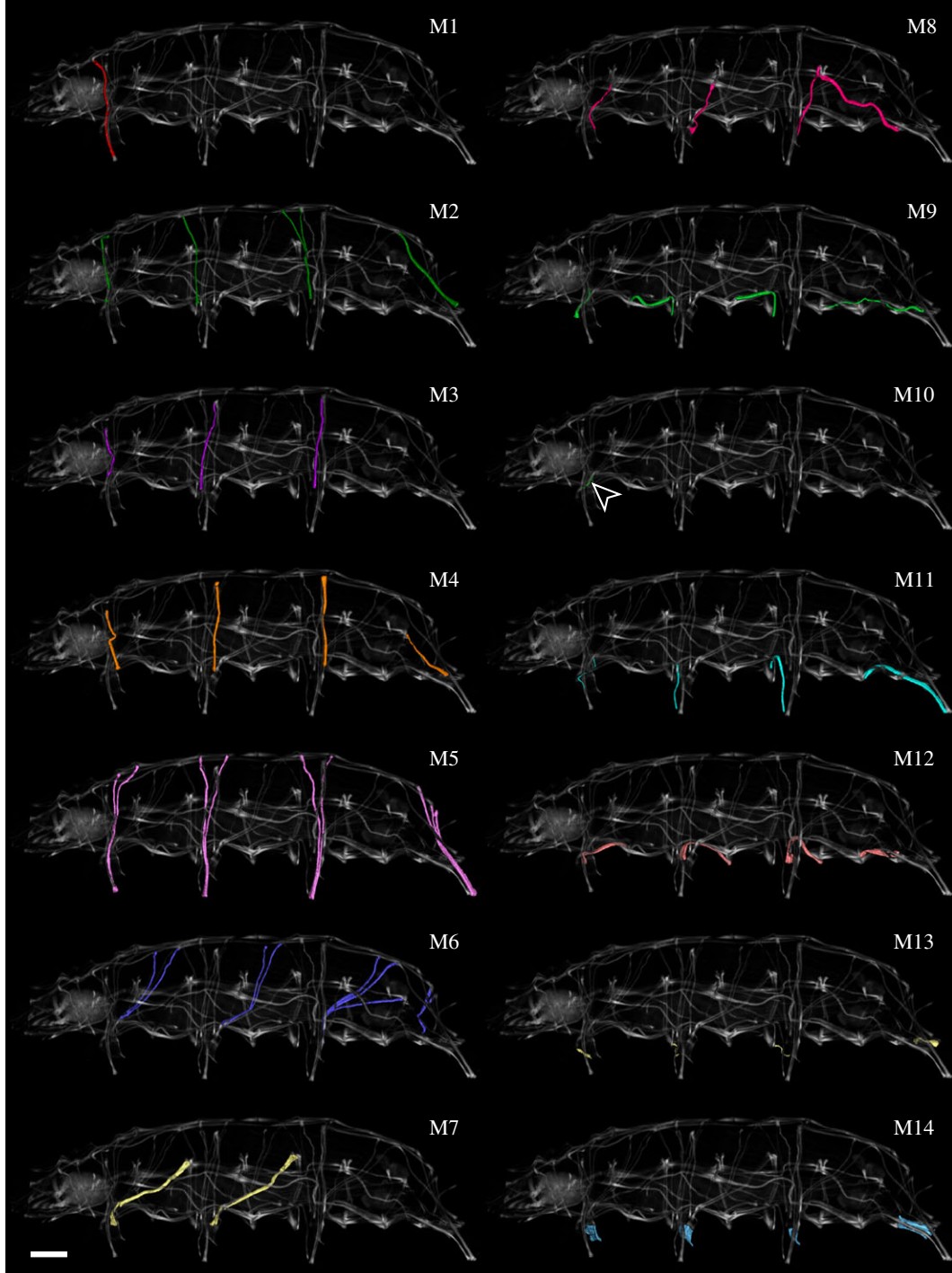

**Figure 3.** Hypothesized serially homologous leg muscles, individually highlighted. Three-dimensional reconstruction based on CLSM data of F-actin labelling. Lateral view; anterior is left, dorsal is up in all images. Scale bar, 20 μm (for all images).

(figure 2). In the following description, we refer to a specific muscle (e.g. iiM2) or, alternatively, to all serial homologues of a muscle by not designating a leg (e.g. M2). Hypotheses of serial homologies are based on (i) the position of all attachment sites of each muscle, both distal (closest to the tip of the leg) and proximal (within or closest to the trunk), (ii) the spatial relationships between muscles and muscle groups, and (iii) cellular morphology of individual muscles, where applicable. Muscles were named in order from anterior to posterior and dorsal to ventral according to their proximal attachment sites, except for muscles of leg IV, which were named according to their serial homologues in legs I–III (figures 2 and 3 and table 1).

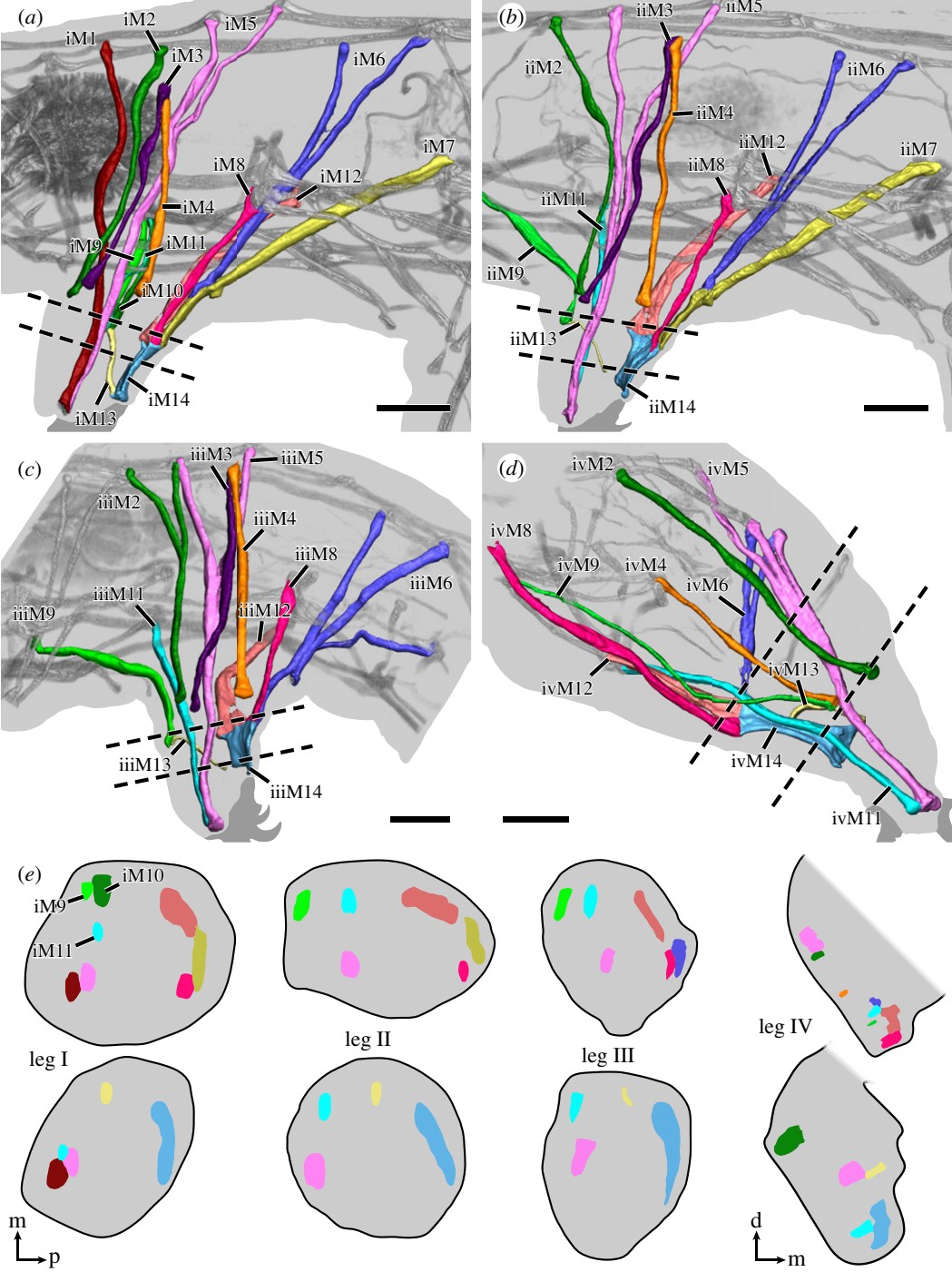

**Figure 4.** Leg muscles of the tardigrade *H. exemplaris*. Three-dimensional reconstructions of legs I–IV (*a–d*, respectively) based on CLSM datasets of F-actin labelling. Dotted lines indicate planes of cross-sections shown in (*e*). Colour-coding according to hypothesized serial homologues. Lateral view; anterior is left, dorsal is up. (*a*) Muscle iM1 (red) does not share a serial homologue in any other leg. (*c*) Muscle iiiM2 (green) shows a branching morphology, unique to leg III. Muscle iiiM6 (blue) consists of three strands, with the posteriormost strand possibly representing a functional analogue of iM7 and iiM7. (*d*) Leg IV is rotated backwards relative to legs I–III so that ivM14 occupies an anteroventral position within the leg. Muscle ivM5 (pink) consists of three strands. (*e*) Schematic diagrams illustrating cross-sections of legs I–IV. Virtual sectioning planes are indicated by dotted lines in (*a–d*); top row corresponds to upper plane, bottom row to lower plane. Note how muscles of legs I–III are concentrated either at the anterior or posterior region of the leg, with muscle M13 spanning the two regions. Muscle M14 occupies the majority of the distal posterior region of legs I–III. Cross-sections not drawn to scale. Orientations in (*e*) for legs I–III (left) or leg IV (right). Orientation legends: d, dorsal; m, medial; p, posterior. Scale bars, 10 µm (*a–d*).

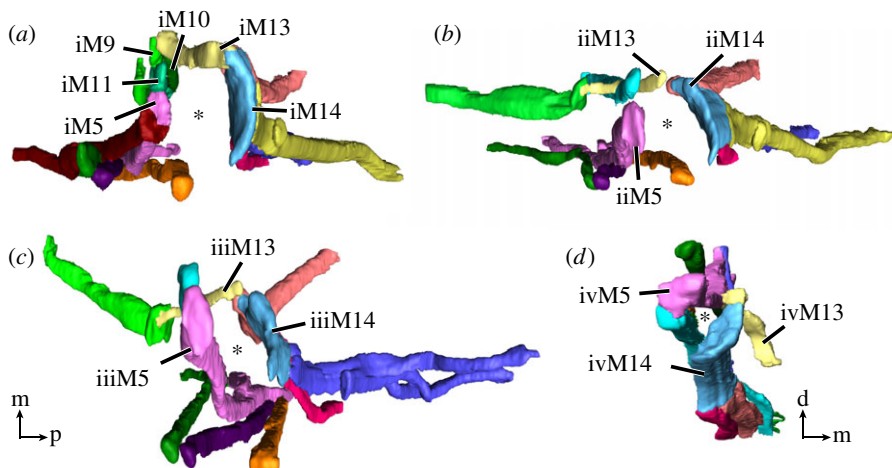

**Figure 5.** Leg muscles of the tardigrade *H. exemplaris* from the distal perspective. Colour-coding according to hypothesized serial homologies. Three-dimensional reconstructions of legs I–IV (*a–d*, respectively) based on CLSM datasets of F-actin labelling. Note how muscles are arranged around the periphery of each leg—primarily in the anterior and posterior regions of each leg—with a large space in the middle (asterisk). The M13 muscles are the only muscles that cross between the anterior and posterior regions. (*d*) The muscles of leg IV are laterally compressed, presenting a smaller empty space between the muscles. Orientation in (*c*) for (*a–c*). Orientation legends: d, dorsal; m, medial; p, posterior.

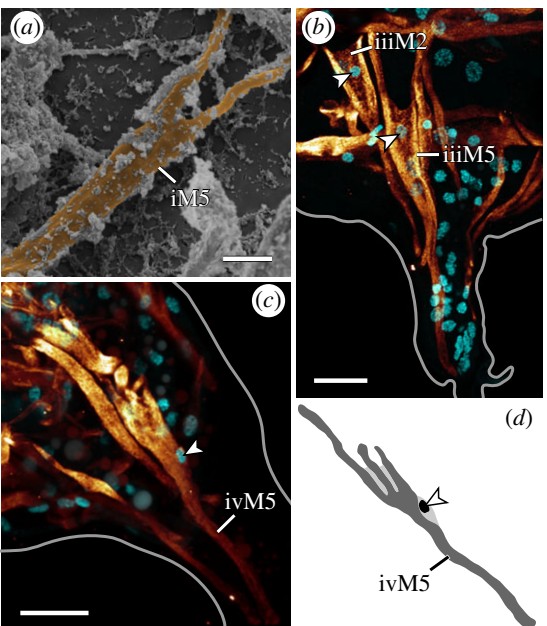

**Figure 6.** Branched morphologies in leg muscles. Lateral views; anterior is left, dorsal is up in all images. (*a*) False-coloured scanning electron micrograph of a bisected specimen showing iM5. (*b,c*) CLSM substacks; anti-myosin II immunolabelling (glow) and nuclear counterstain (cyan). (*b*) Leg III. The branched muscles iiiM2 and iiiM5 with their respective nuclei (arrowheads) located near the point where the two strands of each muscle split. (*c,d*) Muscle ivM5 and its three strands. Note how its nucleus (arrowhead) is located below the point where the three strands split. Scale bars, 3 µm (*a*); 10 µm (*b,c*).

The distal attachment site of each leg muscle can generally be assigned to either the anterior or posterior region of each leg, where muscle attachment sites are concentrated (figure 5*a–d*). Muscle attachment sites exhibit a distinct morphology that allows them to be recognized both externally and internally (figures 8*a–d* and 9). Externally, areas associated with muscle attachment sites exhibit what appear to be a series of tiny holes and, in some cases, a ridge or fold in the cuticle (figures 8*a–d* and 9). Internally, ultrastructural analysis shows that the holes do not form complete channels but rather correspond to invaginations of the cuticle that are associated with filaments from the muscle attachments (figure 10*a–d*). The ridges, on the other hand, correspond to thickenings of the cuticle, more

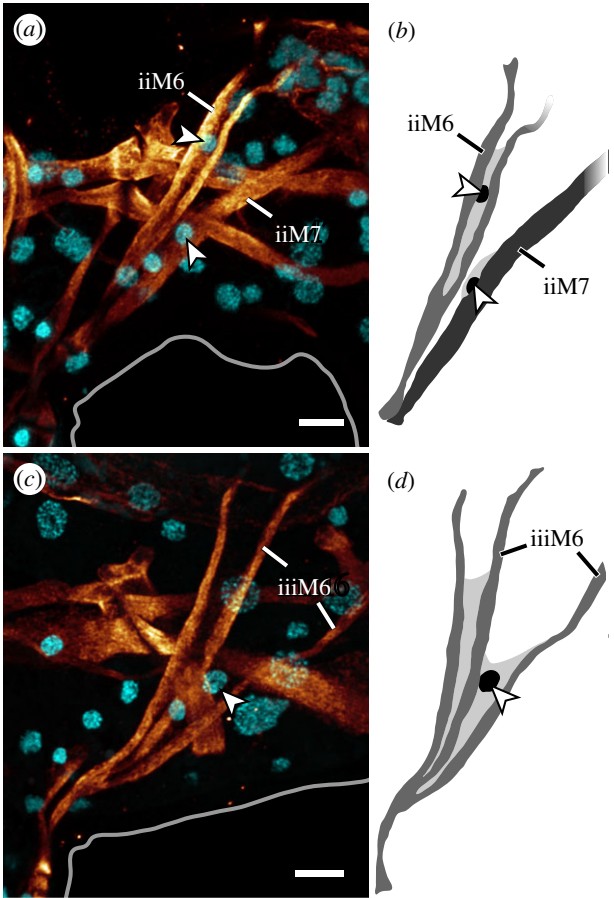

**Figure 7.** Legs II and III showing the relationship between muscles M6 and M7. Lateral views of CLSM substacks; anterior is left, dorsal is up. Anti-myosin II immunolabelling (glow) and nuclear counterstain (cyan). (*a,b*) Muscles iiM6 and iiM7, each with their own nucleus (arrowheads). (*c,d*) Muscle iiiM6, showing three strands and one nucleus (arrowhead). Scale bars, 5 μm (*a–d*).

**Table 1.** Nomenclature and serial homology of individual leg muscles in *H. exemplaris*. Bottom row indicates the total number of muscles in each leg.

| leg I | leg II | leg III | leg IV |
|---|---|---|---|
| iM1 | — | — | — |
| iM2 | iiM2 | iiiM2 | ivM2 |
| iM3 | iiM3 | iiiM3 | — |
| iM4 | iiM4 | iiiM4 | ivM4 |
| iM5 | iiM5 | iiiM5 | ivM5 |
| iM6 | iiM6 | iiiM6 | ivM6 |
| iM7 | iiM7 | — | — |
| iM8 | iiM8 | iiiM8 | ivM8 |
| iM9 | iiM9 | iiiM9 | ivM9 |
| iM10 | — | — | — |
| iM11 | iiM11 | iiiM11 | ivM11 |
| iM12 | iiM12 | iiiM12 | ivM12 |
| iM13 | iiM13 | iiiM13 | ivM13 |
| iM14 | iiM14 | iiiM14 | ivM14 |
| **14** | **12** | **11** | **10** |

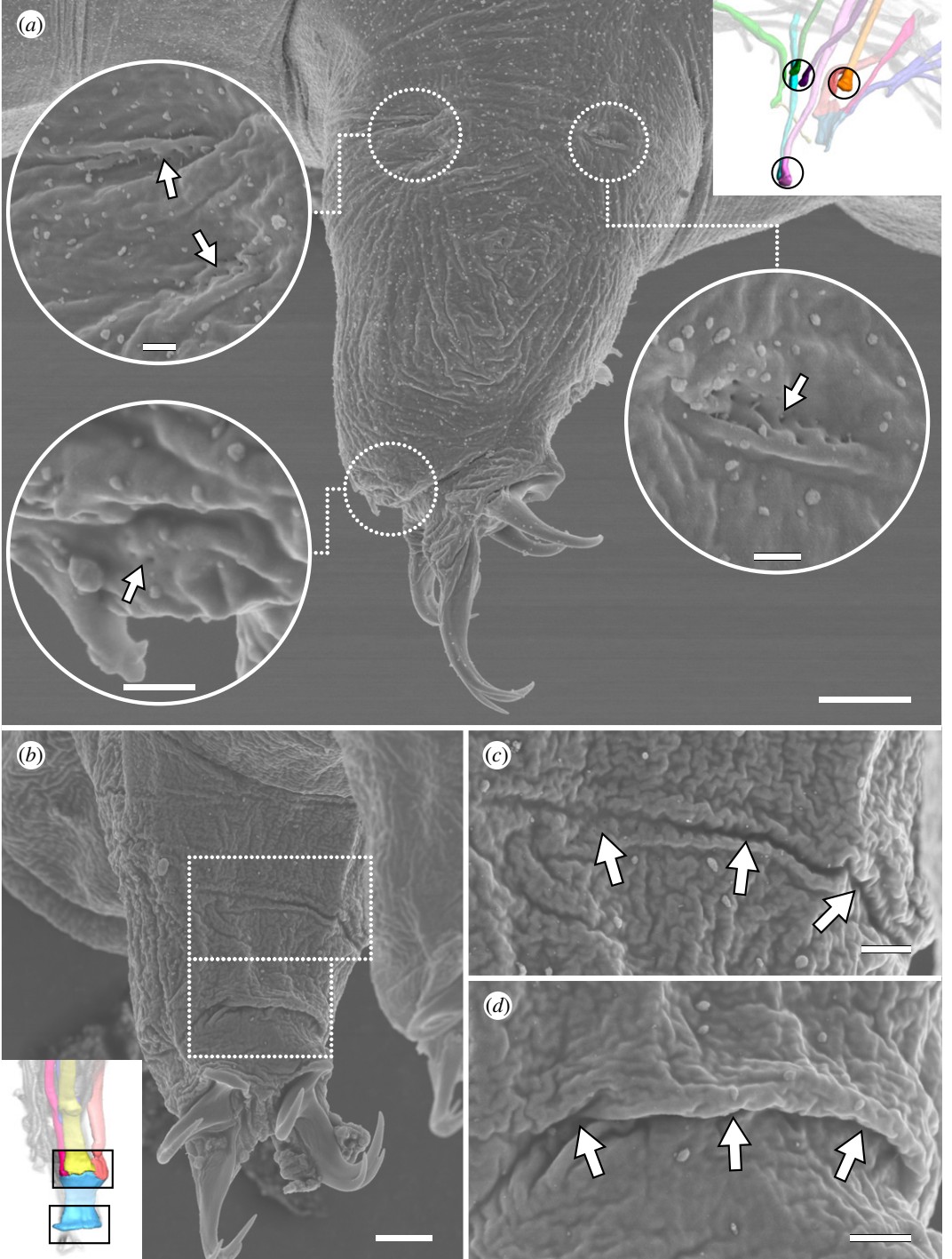

**Figure 8.** Lateral and posterior leg muscle attachment sites seen on the external cuticle. Scanning electron micrographs. Insets (in *a,b*) show three-dimensional reconstructions based on F-actin labelling of the same perspectives with corresponding regions outlined. (*a*) Lateral and distal anterior muscle attachment sites of leg III. Each attachment site (outlined and shown as a close-up) is recognizable by what appear to be tiny holes in the cuticle (arrows) and, in some cases, a ridge or fold. Lateral view; anterior is left, dorsal is up. (*b–d*) Attachment sites of iM14. (*c,d*) shows close-ups of the upper and lower outlined areas in (*b*), respectively. Posterior view of leg I; medial is right, dorsal is up. Note the wide, thick folds of the cuticle at both the proximal (*c*) and distal (*d*) iM14 attachment sites. Scale bars, 5 µm (*a*); 500 nm (insets in *a*); 3 µm (*b*); 1 µm (*c,d*).

specifically to the electron-lucent procuticular layer (*sensu* Shaw [16]; figure 10*a–d*). Otherwise, our data are largely in line with previous studies, which show a similar ultrastructure of somatic muscle attachments among all tardigrade species examined so far [16,18,19].

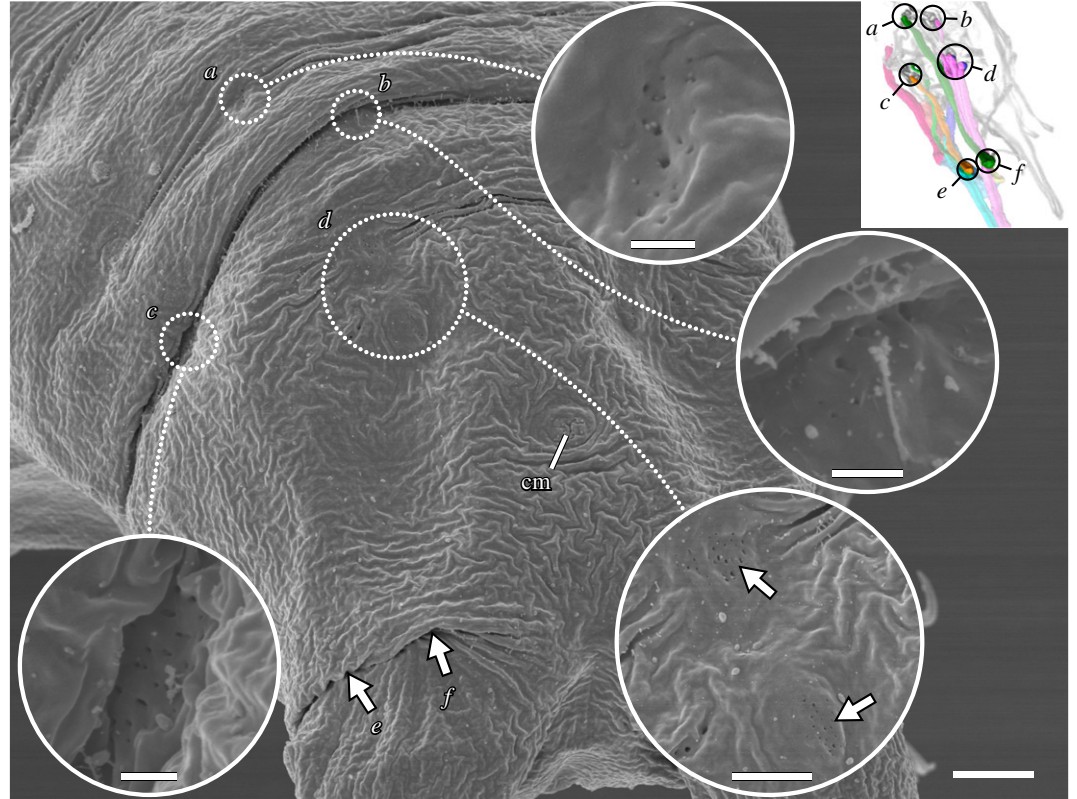

**Figure 9.** Posterior end of the body showing the attachment sites associated with leg IV. Scanning electron micrographs. Inset shows three-dimensional reconstruction based on F-actin labelling of the same perspective with corresponding regions outlined. Posterodorsal view; anterior is upper left. Muscle attachment sites are recognizable by what appear to be tiny holes in the cuticle. Outlined regions of interest (*a–d*) are shown in close-ups. Distal attachment sites (*e,f*) are located under the fold in the leg in this specimen. cm, attachment site of the cloacal muscles. Scale bars, 4 µm (overview); 500 nm (close-ups of *a–c*); 2 µm (close-up of *d*).

## 3.2. Musculature of the legs

In each leg, the muscles are arranged around the periphery of the leg—generally either in the anterior or posterior region of the leg—while the large space in the middle is filled with haemolymph (figures 4*a–e* and 5*a–d*; electronic supplementary material, movie S2). The anterior region of each leg contains a higher number of muscles than the posterior region (figure 4*e*). Our data show that the number of muscles per leg decreases from anterior to posterior, i.e. that leg I has the highest number of leg muscles and leg IV has the lowest number (table 1). Specifically, we identified 14, 12, 11 and 10 muscles in legs I, II, III and IV, respectively (table 1).

Two muscles are unique to leg I, iM1 and iM10, and do not share a serial homologue in any other leg (figures 2–4*a* and table 1). Both belong to the group of leg muscles located in the anterior region of the leg. Muscle iM1 is the anteriormost leg muscle in the body, running laterally next to the pharynx (figure 4*a*). It attaches distally in the anterior tip of the leg and proximally along the dorsal longitudinal musculature. Muscle iM10 runs parallel to iM9, both of which are very short and attach proximally at ventral attachment site 1 (*sensu* Smith & Jockusch [15]), with iM9 attaching on the contralateral side of this attachment site (figure 3; electronic supplementary material, movie S1). Both iM9 and iM10 run very close to and share ventral attachment site 1 with iM11, which attaches distally at the anterior tip of the leg (figures 3, 4*e* and 5*a*). In legs II–IV, each M9 muscle attaches proximally at the anterior ventral attachment site of its respective segment, while M11 attaches at the posterior ventral attachment site of this segment (figures 3 and 4*b–d*). In other words, iiM9, iiiM9 and ivM9 attach at ventral attachment sites 2, 4 and 6, respectively, while iiM11, iiiM11 and ivM11 attach at ventral attachment sites 3, 5 and 7, respectively (figures 2 and 3). Distally, ii–ivM9 attaches at the base and ii–ivM11 at the tip of each leg (figures 2 and 3).

The M2 muscles attach distally at the lateral base of the leg and proximally at the dorsal longitudinal musculature (figures 3 and 8*a*; electronic supplementary material, movie S2). In legs II–IV, they represent

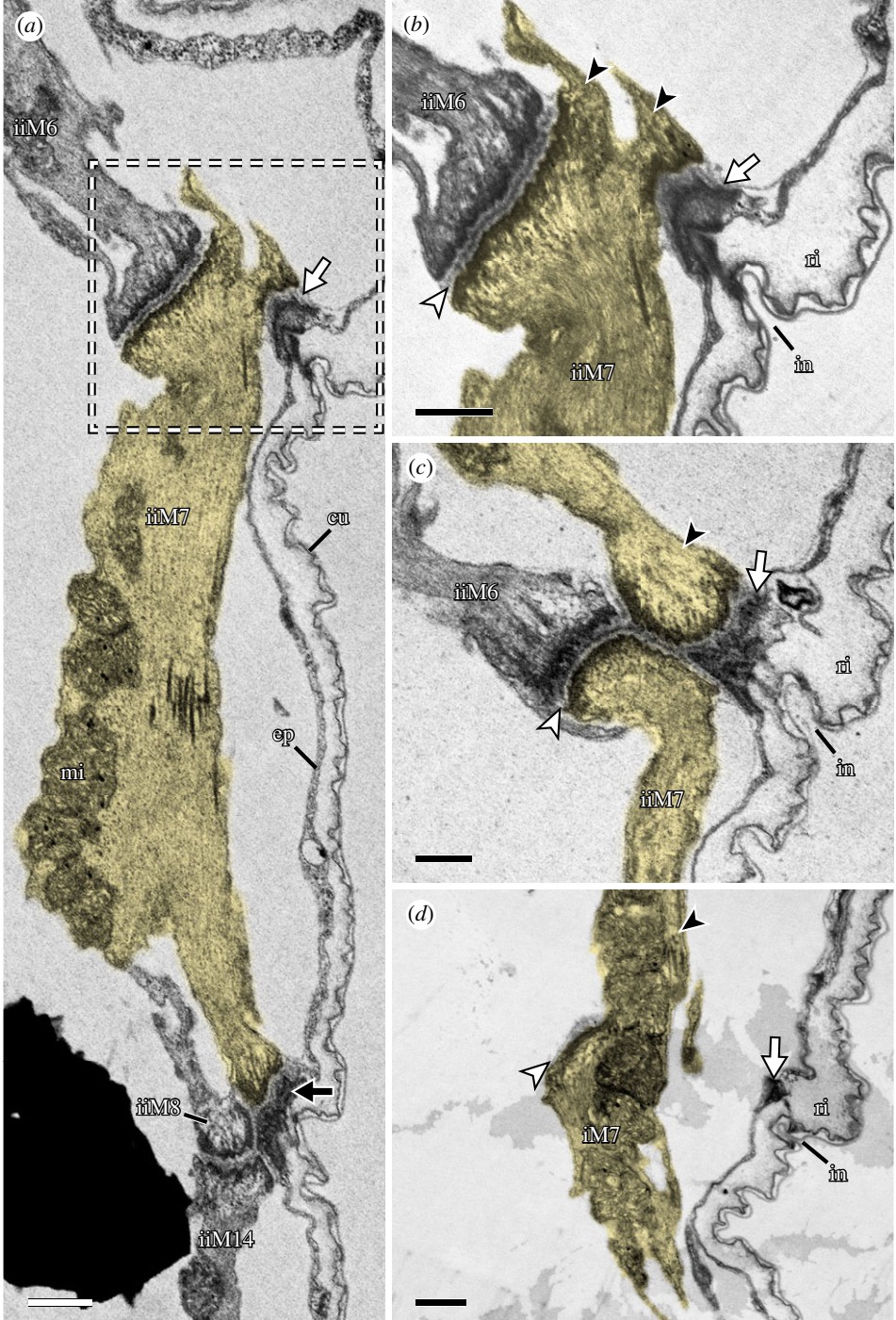

**Figure 10.** Fine structure of muscle M7. STEM-in-SEM images. Sagittal sections; anterior is left, dorsal is up. (*a*) Overview of the posterior base of leg II showing the distal region of muscle iiM7 (false-coloured in yellow) and its intermediate (white arrow) and distal (black arrow) attachment sites. The distal attachment site is shared with iiM14, iiM8 and iiM12 (not shown in this image). (*b–d*) Series of sections through the area outlined in (*a*). White arrowheads indicate the muscle–muscle attachment between M6 and M7. Black arrowheads indicate the beginning of the proximal portion of muscle M7. cu, cuticle; ep, epidermis; in, cuticular invagination; mi, mitochondria; ri, cuticular ridge. Scale bars, 1 µm (*a*); 500 nm (*b–d*).

the anteriormost dorsally attaching leg muscles; iiiM2 is unique in that it shows a bifurcated morphology, branching at approximately half its length (figures 3, 4*b–d* and 6*b*).

Two additional muscle groups, M3 and M4, have distal attachments at approximately the same level as M2, i.e. at the base of the leg, and both attach on the lateral side (figures 3, 4*a–d* and 8*a*; electronic supplementary material, movie S2). Muscles M3 and M4 share a proximal attachment site located at

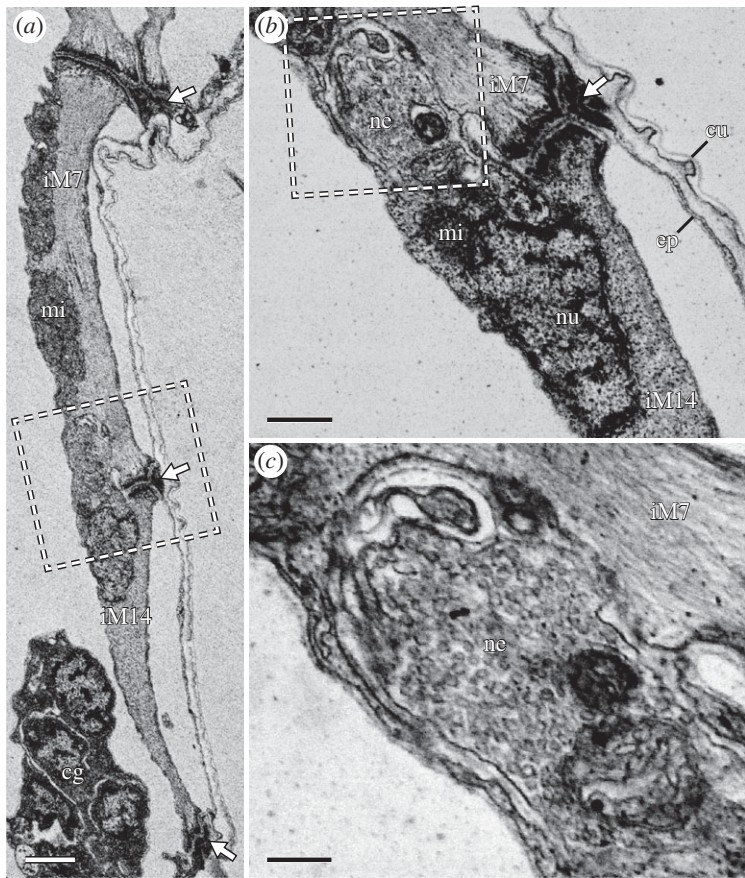

**Figure 11.** Fine structure of iM14, with details of its proximal attachment site. STEM-in-SEM images. Sagittal sections; anterior is left, dorsal is up. White arrows indicate muscle attachment sites. (*a*) Overview of the posterior region of leg I showing iM14 and the distal region of iM7. (*b*) Close-up of the region outlined in (*a*) showing the iM14 proximal attachment site. A neuromuscular junction is adjacent to the attachment site. (*c*) Close-up of the region outlined in (*b*) showing the neuromuscular junction, which is full of synaptic vesicles. cg, claw gland; cu, cuticle; ep, epidermis; mi, mitochondria; ne, neuromuscular junction; nu nucleus. Scale bars, 1 μm (*a*); 500 nm (*b*); 200 nm (*c*).

an intermediate position of the body wall between the lateral and dorsal longitudinal muscle groups (figures 3 and 4*a–d*). The serial homologue of M3 is missing in leg IV (figure 3).

The M5 muscles attach distally at the anterior tip of each leg and branch proximally, with all pairs of proximal attachment sites located along the dorsal longitudinal musculature in legs I–III (figures 3 and 4*a–d*). These muscles are oriented almost perfectly along the dorsoventral axis in legs I–III, i.e. the proximal attachment sites are located directly above the middle of each leg (figures 2, 3 and 4*a–c*). The nucleus of each M5 muscle is located between the two strands, shortly above where the strands split (figure 6*b*). Muscle ivM5 instead splits into three strands proximally, with the nucleus located further distally, before the three strands split (figure 6*c,d*). The two posteriormost proximal attachment sites of ivM5 are shared with those of ivM6 (figure 4*d*).

The M6 muscles branch into two strands in legs I, II and IV, where they attach distally at the posterior base of each leg and proximally along the dorsal longitudinal musculature (figures 3, 4*a–d* and 7*a,b*). In contrast to the M5 muscles, the M6 muscles are not as vertically oriented. Instead, the proximal attachment sites are always located further posteriorly than the distal attachment sites. Muscle iiiM6 is different in that it splits into three branches, with the posteriormost branch attaching proximally along the lateral body wall, where it shares an attachment site with the lateral longitudinal musculature and muscle ivM4 (figures 2, 3, 4*c,d* and 7*c,d*). In addition, the distal attachment site of iiiM6 is situated at approximately half the length of the leg, although this muscle also retains an intermediate attachment site at the base of the leg as in legs I and II (figure 4*c*). The posteriormost strand of iiiM6 corresponds in attachment sites and course to iM7 and iiM7, thereby superficially appearing to be the serial homologue of M7 (figures 3, 4*a–c* and 7*a–d*). However, myosin II and nuclear labelling reveal that this strand together with the other two strands of iiiM6 in fact constitute a single cell (figure 7*c–d*). In all M6 muscles, the nucleus is located after the split and between the strands, specifically between the two posterior strands in iiiM6 (figure 7*a–d*).

The M7 muscles represent the posteriormost-reaching muscles of legs I and II (figures 2, 3 and 4a,b). These muscles are linear in morphology but show three attachment sites: one proximal along the lateral body wall of the posterior adjacent segment, one distal at approximately half the length of the leg and one intermediate site at the posterior base of the leg (figures 4a,b, 10a–d and 11a,b). Each intermediate site of M7 is shared with the distal attachment site of M6, where M7 attaches both to the wall of the leg as well as to M6 (figure 10a–d). Legs III and IV lack a serial homologue of M7 (figure 3).

The distal posterior region of each leg is largely occupied by an M14 muscle; a wide, flattened muscle that follows the curve of the posterior margin of the leg (figures 3, 4a–e and 11a; electronic supplementary material, movie S3). In each M14 muscle, the nucleus and neuromuscular junction are both located near the proximal attachment site (figure 11a–c; electronic supplementary material, movie S3), which is shared with three additional muscles: M7, M8 and M12 in legs I and II or M6, M8 and M12 in legs III and IV (figure 4a–d). Together, this attachment site forms a wide crease on the posterior cuticle of each leg (figure 8b,c). The M8 muscles, which represent the lateralmost muscles of these groups of three, attach proximally along the lateral body wall (figures 2, 3 and 4c,d). Interestingly, iiiM8 and ivM8 share a proximal attachment site (figures 2 and 3). Muscles M12—the muscles of this group that are closest to the midline of the body—attach proximally at ventral attachment sites 2, 4, 6 and 7 in legs I–IV, respectively (figures 2, 3 and 4a–d). Leg IV is different in that muscle ivM12 instead represents the middle muscle of the group, with ivM6 being closest to the midline of the body (figure 4d). Finally, the distal attachment site of each M14 muscle is shared with an M13 muscle, which are short and thin muscles that run anteriorly within the leg to an attachment site shared with an M9 muscle (figures 3 and 4a–d). The M13 and M14 muscles represent true intrinsic muscles, in that they are located in their entirety within the leg.

## 3.3. Remarks on leg IV

The musculature of leg IV is modified considerably compared to that of the anterior three leg pairs, reflecting its derived external morphology. We took the evolutionary rotation of leg IV into account when analysing potential serial homologies. Compared to the other three leg pairs, muscle M14 is more elongate and occupies an anteroventral rather than posterior position within the leg (figures 3 and 4d). Muscles ivM8 and ivM9 are longer than their serial homologues in legs I–III, reflecting the elongate nature of the posterior trunk and posteriormost legs (figures 3 and 4d). The remaining muscles in leg IV, although superficially quite different from those of legs I–III, follow the same relative paths and attach at corresponding attachment sites in the leg and body.

# 4. Discussion

## 4.1. Fine structure and functional implications

Muscle cells with several contractile strands presumably function as a single unit, whereby the contraction of all strands occurs simultaneously. This may be a mechanism to reduce the strain on any single attachment site by distributing the generated force over two or more sites. In onychophorans, for example, some leg muscles are organized as a network of thin fibres and spread out over a relatively large surface in order to minimize the physical deformation of the leg during muscle contraction [34]. The same principle may apply to branched muscles in the tardigrade leg, albeit on the scale of only two or three fibres. A similar branching of individual muscle cells is also known from other metazoans, such as in the caudal spine of nematomorph larvae [35] and the pharynx of a ctenophore [36]. Based on the positions of muscles exhibiting specialized morphologies—which we define as any shape other than the linear strap-shape of somatic muscles—and their presumptive roles in locomotion, we hypothesize that derived muscle morphologies in tardigrade legs are the result of functional requirements above all [37], rather than e.g. evolutionary constraints [38]. This hypothesis is in line with the conclusions of Walz [37] regarding somatic and visceral muscle types in tardigrades, and of Heumann & Zebe [39] regarding the distribution of obliquely striated muscles across annelids, molluscs and nematodes. On the other hand, the morphology of attachment sites and the layout of the muscular system have been shown to be phylogenetically informative within Tardigrada, at least at higher taxonomic levels [14].

The muscle attachments themselves superficially resemble the 'double hemidesmosome' structures first described from rhesus monkey epithelial tissue by Listgarten [19,40]. Interestingly, such structures

have also been described from the cycliophoran *Symbion pandora* [41]. It is important to note that desmosomes and hemidesmosomes are typically associated with cytoplasmic intermediate filaments, which are, however, missing in panarthropods [26]. In *H. exemplaris*, a lamin-derived cytoplasmic intermediate filament (cytotardin) has been co-opted for this task and appears as a belt around the cell, but it occurs exclusively in ectodermal tissues such as the epidermis and foregut [26]. Which protein fills this role in mesodermal tissue is not entirely clear, but our and previous ultrastructural data (see fig. 2 in Shaw [16]) suggest that the muscle attachments of tardigrades are associated with actin rather than intermediate filaments, at least on the side of the muscle cell. In this regard, tardigrade muscle attachments show a unique combination of features: an association with actin filaments like in a typical adherens junction, while a layer of dense material between cells (presumably the basal lamina) resembles the extracellular space of a double hemidesmosome (present study; [19]).

Our hypothesis regarding functional constraints does not only apply to the branched muscles. The position of iM1 as the anteriormost muscle in leg I, for example, may explain why leg I can be extended much further forward than any other leg [42], considering that there are no serial homologues of M1 in legs II–IV. At the posterior of each leg, the morphology of the M14 muscle (the 'sheet muscles' *sensu* Smith & Jockusch [15]) implicates its role in a coordinated contraction of the leg or retraction of the claws, either during locomotion or tun formation. The shape of the M14 attachment sites, i.e. wide horizontal creases along the posterior region of legs I–III may serve to better control the plane in which each leg folds and prevent its collapse toward a single small point during muscle contraction. The anteroventral position of ivM14 may also facilitate the grasping action that has been proposed to be one of the main functions of leg IV [42]. Similarly, the M7 muscles may be responsible for the retraction of legs I and II, and the presence of an intermediate attachment site along the length of each of these muscles may prevent kinks in the base of the leg during contraction. It should be noted that in leg III, the M7 homologue may have been replaced by the posteriormost strand of the triple-branched iiiM6 both in form and, presumably, in function, as it shows the same relative path and position of attachment sites as the M7 muscles. Alternatively, an ancestral branched muscle may have split into separate cells in legs I and II for more precise control of individual muscles. We favour the first scenario, as a higher number of leg muscles are seen in the other extant lobopodians, i.e. the onychophorans [34], and a reduction might have occurred in tardigrades at several levels due to miniaturization [43]. Tardigrade leg musculature is reduced not only in the number of muscles but also in complexity compared to that of onychophorans. For example, although most muscles are arranged around the periphery of the leg in both taxa, onychophorans possess additional leg muscles that partition the leg into compartments, as well as twisted muscles that are responsible for the rotation of the leg [34]. The lack of the latter in tardigrades partially explains the simple back-and-forth movement of the legs (without rotation) characteristic of tardigrade locomotion [42]. Whatever the case may be, we hypothesize that a reduced number but higher degree of branching of muscles towards the posterior of the animal (e.g. iiiM2, iiiM6 and ivM5) may be related to the observation that movement of the posterior legs is not as precise during locomotion as that of the anterior two leg pairs [42]. A comparison of leg musculature between tardigrade species with different gaits is necessary to test this hypothesis.

## 4.2. Comparison of myoanatomy with previous studies

Several previous studies have mapped out the muscular system of different tardigrade species. Investigations of the musculature of *H. exemplaris* or closely related species [10,11,14,15] generally agree on the layout of attachment sites with little deviation (see electronic supplementary material, table S1 for a comparison). While our data are largely in line with previous studies in terms of the positions of muscle attachment sites, the leg muscles themselves are more difficult to compare. This challenge partly comes from the fact that the nomenclature traditionally used in the literature is mainly focused on attachment sites and does not always account for the fact that branched leg muscles have more than two attachment sites. Müller [10] was able to identify most of the branched leg muscles but named each strand according to its attachment sites regardless. Her depiction of the sheet muscle (iiM14 herein) and the arrangement of muscles associated with its proximal attachment site (iiM7, iiM8 and iiM12 herein) largely mirror our results (compare with fig. 4b in [10]). The exception is that our data show that the split between muscles '11' and 'i' illustrated by Müller [10] actually corresponds to the distal attachment site of iiM6 and the intermediate attachment site of iiM7, respectively. In this case, this disparity might be attributable to species-specific differences, as the illustration of Müller [10] is based on *Macrobiotus*

*hufelandi*. In fact, Müller [10] notes that even the closely related *Hypsibius convergens* and *Ramazzottius oberhaueseri* show differences in the number and grouping of appendage muscles.

More recent studies based on fluorescent labelling of F-actin in *H. exemplaris* [15] and *Hypsibius* sp. [11] generally differ only in details of individual muscles. Although the musculature of leg IV is the greatest point of disagreement, previous studies still consistently show that leg IV contains the fewest muscles, with a maximum total of seven [10,11,15]. This notable deviation from the 10 muscles we describe herein may be due to a number of factors. It is possible that the super-resolution CLSM module used for the present study allowed us to identify muscles or parts of muscles that were previously overlooked using conventional techniques, for example, muscle ivM13. However, ivM13, the smallest and thinnest muscle in leg IV, was already described by Müller [10] in *R. oberhaueseri* (muscle 'P$_1$–an' in fig. 5 in [10]) and is also visible in preparations of other species, including *Hypsibius* sp. (cf. fig. 3a in [11]), *Paramacrobiotus richtersi*, *Milnesium* cf. *tardigradum*, *Acutuncus antarcticus* (cf. figs 6 and 7 in [14]), and the previous investigation of *H. exemplaris* (cf. fig. 7C in [15]). Therefore, while the improved resolution might uncover fine details that may help to inform serial homologies, it did not reveal any muscles that were previously undiscovered. Ironically, the non-contractile region of a branched muscle cell that spans the two contractile strands is invisible to phalloidin labelling, which may partly explain why our combined approach using anti-myosin II immunolabelling revealed details that in some cases more closely match earlier studies based on bright-field observations (e.g. [10]) rather than more recent F-actin-based investigations (e.g. [11,15]).

We find it more plausible that at least some differences can be attributed to a matter of interpretation, given that our description of the sheet muscles (M14) and muscles associated with their proximal attachment sites more closely matches that of the more distantly related *H. crispae* [12] than previous investigations of *H. exemplaris* [15] or *Hypsibius* sp. [11]. Specifically regarding leg IV, we believe that the major difference in the number of described muscles between the present and previous studies can be ascribed to the unique morphology of the leg itself. Leg IV faces backwards and essentially appears as a posterior extension of the trunk, complicating the distinction between the leg itself and the posteriormost trunk segment. This modification relative to legs I–III is especially evident in the sheet muscle (ivM14), which is not only longer than its serial homologues in legs I–III but is also aligned more with the anterior–posterior body axis rather than the dorsoventral body axis as in the other legs. If leg IV does indeed possess rotated serial homologues of most of the muscles found in legs I–III and this pattern is common to other tardigrade species, it could explain why its movement in *Echiniscus testudo* and *Macrobiotus* sp. was described as similar to but opposite of legs I–III ('ähnlich, aber entgegengesetzt,' [42]). Taking these modifications into account and viewing the muscles in this light lead us to the conclusion that the musculature of leg IV is, though slightly reduced, less derived than previously thought [14].

# 5. Conclusion

The results presented herein show that virtually all leg muscles are serially repeated, adding further support for the hypothesis that tardigrade musculature is largely a metamerically arranged organ system [14,15,21]. The striking reduction in the total number of leg muscles from anterior to posterior combined with an increased degree of branching begs the question of how such an arrangement evolved. It is known that in *H. exemplaris*, the segmental trunk ganglia arise from anterior to posterior during development [44]. It is possible that the musculature follows the same sequence, in which case heterochronic shifts in development might be responsible for the observed differences in musculature between the different legs. Along these lines, is increased branching of muscle strands a functional compensation for the reduction in the number of muscle cells, or perhaps vice versa? Studies of embryonic and postembryonic development may help address these questions by documenting how and when each muscle arises, particularly regarding the branched muscles. It has been shown previously that newly hatched juveniles lack the tiny cuticular pores indicative of muscle attachment sites [45], suggesting that the cuticle or musculature might continue developing to some extent postembryonically. Do the underlying muscles in juveniles also differ morphologically from those in adults? Finally, comparative studies of the terminal or otherwise highly derived segments of onychophorans or fossil lobopodians may provide further insights into the nature of serial homologies in these animals (e.g. [46]).

Ethics. The experiments in this study did not require approval by an ethical committee. All procedures in this investigation complied with international and institutional guidelines, including the guidelines for animal welfare as laid down by the German Research Foundation (DFG).

Data accessibility. The datasets supporting the conclusions of this article are included within the article and its additional files. The raw CLSM stacks together with the label fields that were used to generate the three-dimensional reconstructions are available within the Dryad Digital Repository: https://doi.org/10.5061/dryad.kv1cr18 [47].

Authors' contributions. Conception and design: V.G. and G.M. Acquisition of data: V.G. Analysis and interpretation of data: V.G. and G.M. Drafting of the manuscript: V.G. Revision of drafts: V.G. and G.M. Final approval of submitted version: V.G. and G.M.

Competing interests. We declare we have no competing interests.

Funding. We received no funding for this study.

Acknowledgements. Sonja Fuhrmann and Sonja Kasten (University of Kassel) are acknowledged for assistance with western blotting. The authors thank Ivo de Sena Oliveira (University of Kassel) and Sarah Atherton (Swedish Museum of Natural History) for constructive comments and critical reading of the manuscript. We thank Markus Maniak and Heike Otto (University of Kassel) for preparing and providing the hybridoma supernatant. We are thankful to two anonymous reviewers, whose constructive comments greatly helped to improve the manuscript.

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
