## [Reviewer comments · Royal Society Open Science]

Review History

RSOS-191159.R0 (Original submission)

Review form: Reviewer 1

Is the manuscript scientifically sound in its present form?

Yes

Are the interpretations and conclusions justified by the results?

Yes

Is the language acceptable?

Yes

Do you have any ethical concerns with this paper?

No

Have you any concerns about statistical analyses in this paper?

No

Recommendation?

Accept with minor revision (please list in comments)

Comments to the Author(s)

Summary: In this manuscript, the authors perform a very detailed and rigorous analysis of the leg musculature of the eutardigrade *Hypsibius exemplaris*. The analysis focuses on a very important problem that has not been adequately addressed in previous studies – identifying serially homologous muscles between the segmentally reiterated legs of a tardigrade. This study is all the more important, given that *H. exemplaris* has recently become the developmental model for evolutionary studies of tardigrades. This study provides a much-needed guide for recognizing particular segmental identities, the developmental basis of which may be unraveled in future developmental studies. Additionally, this study clearly demonstrates that intrinsic muscles are found in the legs of tardigrades, which may lead to a fairly large shift in our understanding of the transition of lobopodal legs to the jointed appendages characteristic of arthropods. The data and methods used to collect it are of very high quality, and the text is well written. This manuscript will be broadly interesting to biologists from several fields that are interested in the evolution or functional morphology of animal legs. I have a few comments that I hope that the authors find useful.

Page numbers referenced below are in the very top left or right corners of the proof. Line numbers referenced below are on the left side of the proof.

Major comments:

1.) Section 4.2, page 6 of 22, lines ~34–36: In reference to the proximal attachment sites of M9 and M11, it is unclear what criterion is being used to delineate segment boundaries. I recommend that the authors explain how they determined that the proximal attachment sites for these muscles were in an anteriorly adjacent segment, and not in the anterior part of the same segment.

2.) Figures 6 and 7, and associated text. Statements in text depend on accurately delineating muscle strands M6 and M7 as separate cells. This is done by counting nuclei – with 1 nuclei expected for each muscle cell. However, it is unclear how the authors determined which nuclei belong to which muscle strand (s). In the images they have arrowheads pointing to particular nuclei, but several other nuclei are also found in close proximity to the muscle fibers in question. I recommend that the authors explain how they identified the nuclei of particular muscles strands.

Minor comments:

Section 2, page 3 of 22, line 48: The anteriormost ventral attachment site in *H. exemplaris* sits immediately behind the first trunk ganglion, so it is not in the head, as stated. I recommend modifying this statement.

Section 4.1, page 6 of 22, line ~8: Refers to figure 1a–d, but figure 1 only includes panels a and b. Please fix this.

Section 4.2, page 6 of 22, line ~44: Figure 8a–d is referred to as evidence of a large internal haemolymph in the legs, but this figure primary shows SEMs of the external cuticle. I recommend deleting reference to this figure in this regard.

Section 4.2, page 6 of 22, line ~54: Referring to “(figure 4a, e). This reference seems to be to support the statement that iM11 attaches distally. However, iM11 is not labeled in figure 4a. I

recommend labeling it, or rewriting the sentence in question to make it more clear what the reference to “(figure 4a, e)” is meant to support.

Section 4.2, page 6 of 22, line ~56–bottom of page: Referring to lateral position within the legs of muscles M2, M3, and M4. Whether this attachment site is at the lateral part of the leg is difficult to determine based on the figures in text. The best demonstration of this statement is found in the inset model found in figure 8a. I recommend referencing this figure as evidence for the lateral location of the distal attachment sites of these muscles. Also, I recommend referencing any of the supplemental movies that show the distal location of these attachment sites.

Section 4.2, page 6 of 22, last paragraph: Referring to muscles M3 and M4. The text states that the distal attachment sites of these muscles are at “approximately half the length of the leg”, but they appear to attach in the proximal most part of the leg, much more proximal than stated. I recommend modifying the text to make this clear.

Section 4.2, page 7 of 22, ~lines 3–4: Referring to the statement that the “nucleus of each M3 muscle is located between the two strands..”, with the associated reference to figure 6b. Figure 6b shows that the nuclei of both M2 and M5 are located between the split in these strands, but M3 is not split, and is not labeled in this panel. I suspect that the authors meant to refer to muscles M2 and M5, not M3. If so, I recommend making this change in text. If not, please modify the figure panel to show where muscle strand M3 is located.

Section 4.2, page 7 of 22, ~lines 11–13: Referring to “(figure 7a–d)” as evidence that the “posteriormost strand of iiiM6 corresponds in attachment sites and course to iM7 and iiM7...”. I recommend also referencing “(Figure 3, 4a–c)”.

Section 4.2, page 7 of 22, ~line 19–20: Referring to shared attachment sites between M7 and M6. The figure reference should be “(figure 10a–c)”, not “(figure 10a–d)”, since M6 is not shown in panel d. I recommend making this change.

Review form: Reviewer 2

Is the manuscript scientifically sound in its present form?

Yes

Are the interpretations and conclusions justified by the results?

Yes

Is the language acceptable?

Yes

Do you have any ethical concerns with this paper?

No

Have you any concerns about statistical analyses in this paper?

No

Recommendation?

Major revision is needed (please make suggestions in comments)

Comments to the Author(s)

General comment:

Aiming at understanding if tardigrade leg muscles are serially repeated, the authors investigated the overall arrangement as well as ultrastructural aspects of the leg musculature of *Hypsibius exemplaris*. In order to achieve their goals, authors used an approach based on (immuno)cytochemistry tools and advanced microscopy techniques.

All confocal laser scanning and electron micrographs and 3D reconstructions presented in the figure plates are of high quality, which certainly denotes the great effort made by the authors to perform this investigation. Great job!

The introduction to the questions about segmental patterns in Tardigrada, especially in what concerns leg myoanatomy, is well exposed and the methodology used is well described. However, the description of the findings is in general acceptable but needs to be revamped. Therefore, I recommend the authors to revise the ms. (with an extra care in Results) if its overall quality is to be increased. I certainly look forward to see this ms. published and the novel findings it encloses available to the scientific community.

General comments for the sections:

- Abstract: succinct and the questions that drove the authors are clearly defined. However, it lacks information about the conclusions. Please consider including the main results and conclusions from your study.
- Introduction + Material and Methods: well presented; I made some corrections/comments/suggestions directly on the submission PDF.
- Results: as compared to the rest of the ms., this section clearly lacks quality mainly due to an incomplete, vague description of the findings. As pointed out in several comments embedded in the PDF, it is important to provide the reader with a clear description of the arrangement of all muscles mentioned.
- Discussion: I find subsection 5.1 somewhat speculative in general and made a couple comments about it.
- Conclusions: well presented; it wraps up the whole story coherently.

Major & Minor remarks:

Several comments, corrections and suggestions have been annotated directly (and going straight to the point) on the submitted PDF. These remarks concern the text (including figure captions) and figures (Appendix A).

Decision letter (RSOS-191159.R0)

16-Sep-2019

Dear Mr Gross,

On behalf of the Editors, I am pleased to inform you that your Manuscript RSOS-191159 entitled "Cellular morphology of leg musculature in the water bear *Hypsibius exemplaris* (Tardigrada) unravels serial homologies" has been accepted for publication in Royal Society Open Science subject to minor revision in accordance with the referee suggestions. Please find the referees' comments at the end of this email.

The reviewers and handling editors have recommended publication, but also suggest some minor revisions to your manuscript. Therefore, I invite you to respond to the comments and revise your manuscript.

- Ethics statement

- Data accessibility

<http://datadryad.org/submit?journalID=RSOS&manu=RSOS-191159>

- Competing interests

- Authors' contributions

- Acknowledgements

- Funding statement

Because the schedule for publication is very tight, it is a condition of publication that you submit the revised version of your manuscript before 25-Sep-2019. Please note that the revision deadline will expire at 00.00am on this date. If you do not think you will be able to meet this date please let me know immediately.

Kind regards,
Lianne Parkhouse
Royal Society Open Science
openscience@royalsociety.org

on behalf of Dr Michael Doube (Associate Editor) and Kevin Padian (Subject Editor)
openscience@royalsociety.org

Associate Editor Comments to Author (Dr Michael Doube):

Two reviewers have seen this manuscript and are enthusiastic about its broad implications and the high quality of the images. They both make several detailed suggestions for improvements to the text of the manuscript, which you should endeavour to incorporate in a revision.

Reviewer comments to Author:

Reviewer: 1

Summary: In this manuscript, the authors perform a very detailed and rigorous analysis of the leg musculature of the eutardigrade *Hypsibius exemplaris*. The analysis focuses on a very important problem that has not been adequately addressed in previous studies – identifying serially homologous muscles between the segmentally reiterated legs of a tardigrade. This study is all the more important, given that *H. exemplaris* has recently become the developmental model for evolutionary studies of tardigrades. This study provides a much-needed guide for recognizing particular segmental identities, the developmental basis of which may be unraveled in future developmental studies. Additionally, this study clearly demonstrates that intrinsic muscles are found in the legs of tardigrades, which may lead to a fairly large shift in our understanding of the transition of lobopodal legs to the jointed appendages characteristic of arthropods. The data and methods used to collect it are of very high quality, and the text is well written. This manuscript will be broadly interesting to biologists from several fields that are interested in the evolution or functional morphology of animal legs. I have a few comments that I hope that the authors find useful.

Page numbers referenced below are in the very top left or right corners of the proof. Line numbers referenced below are on the left side of the proof.

Major comments:

1.) Section 4.2, page 6 of 22, lines ~34–36: In reference to the proximal attachment sites of M9 and M11, it is unclear what criterion is being used to delineate segment boundaries. I recommend that the authors explain how they determined that the proximal attachment sites for these muscles were in an anteriorly adjacent segment, and not in the anterior part of the same segment.

2.) Figures 6 and 7, and associated text. Statements in text depend on accurately delineating muscle strands M6 and M7 as separate cells. This is done by counting nuclei – with 1 nuclei expected for each muscle cell. However, it is unclear how the authors determined which nuclei belong to which muscle strand (s). In the images they have arrowheads pointing to particular nuclei, but several other nuclei are also found in close proximity to the muscle fibers in question. I recommend that the authors explain how they identified the nuclei of particular muscles strands.

Minor comments:

Section 2, page 3 of 22, line 48: The anteriormost ventral attachment site in *H. exemplaris* sits immediately behind the first trunk ganglion, so it is not in the head, as stated. I recommend modifying this statement.

Section 4.1, page 6 of 22, line ~8: Refers to figure 1a–d, but figure 1 only includes panels a and b. Please fix this.

Section 4.2, page 6 of 22, line ~44: Figure 8a–d is referred to as evidence of a large internal haemolymph in the legs, but this figure primary shows SEMs of the external cuticle. I recommend deleting reference to this figure in this regard.

Section 4.2, page 6 of 22, line ~54: Referring to “(figure 4a, e). This reference seems to be to support the statement that iM11 attaches distally. However, iM11 is not labeled in figure 4a. I recommend labeling it, or rewriting the sentence in question to make it more clear what the reference to “(figure 4a, e)” is meant to support.

Section 4.2, page 6 of 22, line ~56–bottom of page: Referring to lateral position within the legs of muscles M2, M3, and M4. Whether this attachment site is at the lateral part of the leg is difficult to determine based on the figures in text. The best demonstration of this statement is found in the inset model found in figure 8a. I recommend referencing this figure as evidence for the lateral location of the distal attachment sites of these muscles. Also, I recommend referencing any of the supplemental movies that show the distal location of these attachment sites.

Section 4.2, page 6 of 22, last paragraph: Referring to muscles M3 and M4. The text states that the distal attachment sites of these muscles are at “approximately half the length of the leg”, but they appear to attach in the proximal most part of the leg, much more proximal than stated. I recommend modifying the text to make this clear.

Section 4.2, page 7 of 22, ~lines 3–4: Referring to the statement that the “nucleus of each M3 muscle is located between the two strands..”, with the associated reference to figure 6b. Figure 6b shows that the nuclei of both M2 and M5 are located between the split in these strands, but M3 is not split, and is not labeled in this panel. I suspect that the authors meant to refer to muscles M2

and M5, not M3. If so, I recommend making this change in text. If not, please modify the figure panel to show where muscle strand M3 is located.

Section 4.2, page 7 of 22, ~lines 11-13: Referring to “(figure 7a-d)” as evidence that the “posteriormost strand of iiiM6 corresponds in attachment sites and course to iM7 and iiM7...”. I recommend also referencing “(Figure 3, 4a-c)”.

Section 4.2, page 7 of 22, ~line 19-20: Referring to shared attachment sites between M7 and M6. The figure reference should be “(figure 10a-c)”, not “(figure 10a-d)”, since M6 is not shown in panel d. I recommend making this change.

Reviewer: 2

General comment:

Aiming at understanding if tardigrade leg muscles are serially repeated, the authors investigated the overall arrangement as well as ultrastructural aspects of the leg musculature of *Hypsibius exemplaris*. In order to achieve their goals, authors used an approach based on (immuno)cytochemistry tools and advanced microscopy techniques.

All confocal laser scanning and electron micrographs and 3D reconstructions presented in the figure plates are of high quality, which certainly denotes the great effort made by the authors to perform this investigation. Great job!

The introduction to the questions about segmental patterns in Tardigrada, especially in what concerns leg myoanatomy, is well exposed and the methodology used is well described. However, the description of the findings is in general acceptable but needs to be revamped. Therefore, I recommend the authors to revise the ms. (with an extra care in Results) if its overall quality is to be increased. I certainly look forward to see this ms. published and the novel findings it encloses available to the scientific community.

General comments for the sections:

-Abstract: succinct and the questions that drove the authors are clearly defined. However, it lacks information about the conclusions. Please consider including the main results and conclusions from your study.

- Introduction + Material and Methods: well presented; I made some corrections/comments/suggestions directly on the submission PDF.

- Results: as compared to the rest of the ms., this section clearly lacks quality mainly due to an incomplete, vague description of the findings. As pointed out in several comments embedded in the PDF, it is important to provide the reader with a clear description of the arrangement of all muscles mentioned.

- Discussion: I find subsection 5.1 somewhat speculative in general and made a couple comments about it.

- Conclusions: well presented; it wraps up the whole story coherently.

Major & Minor remarks:

Several comments, corrections and suggestions have been annotated directly (and going straight to the point) on the submitted PDF. These remarks concern the text (including figure captions) and figures.

Author's Response to Decision Letter for (RSOS-191159.R0)

See Appendix B.

Decision letter (RSOS-191159.R1)

23-Sep-2019

Dear Mr Gross,

I am pleased to inform you that your manuscript entitled "Cellular morphology of leg musculature in the water bear *Hypsibius exemplaris* (Tardigrada) unravels serial homologies" is now accepted for publication in Royal Society Open Science.

on behalf of Dr Michael Doube (Associate Editor) and Kevin Padian (Subject Editor)
openscience@royalsociety.org

Appendix A**ROYAL SOCIETY
OPEN SCIENCE****Cellular morphology of leg musculature in the water bear
Hypsibius exemplaris (Tardigrada) unravels serial
homologies**

Journal:	Royal Society Open Science
Manuscript ID	RSOS-191159
Article Type:	Research
Date Submitted by the Author:	04-Jul-2019
Complete List of Authors:	Gross, Vladimir; University of Kassel, Department of Zoology Mayer, Georg; Universitat Kassel Institut fur Biologie, Department of Zoology
Subject:	cellular biology < BIOLOGY, evolution < BIOLOGY
Keywords:	3D reconstruction, F-actin, myosin, functional morphology, lobopod, Ecdysozoa
Subject Category:	Biology (whole organism)

Author-supplied statements

Relevant information will appear here if provided.

Ethics

Does your article include research that required ethical approval or permits?:

This article does not present research with ethical considerations

Statement (if applicable):

CUST_IF_YES_ETHICS :No data available.

Data

It is a condition of publication that data, code and materials supporting your paper are made publicly available. Does your paper present new data?:

Yes

Statement (if applicable):

The datasets supporting the conclusions of this article are included within the article and its additional files.

Conflict of interest

I/We declare we have no competing interests

Statement (if applicable):

CUST_STATE_CONFLICT :No data available.

Authors' contributions

This paper has multiple authors and our individual contributions were as below

Statement (if applicable):

Conception and design: VG and GM. Acquisition of data: VG. Analysis and interpretation of data: VG and GM. Drafting of manuscript: VG. Revision of drafts: VG and GM. Final approval of submitted version: VG and GM.

Cellular morphology of leg musculature in the water bear *Hypsibius exemplaris* (Tardigrada) unravels serial homologies

Vladimir Gross^{1,*} and Georg Mayer¹

¹Department of Zoology, Institute of Biology, University of Kassel, Heinrich-Plett-Straße 40, D-34132 Kassel

Keywords: 3D reconstruction, F-actin, myosin, functional morphology, lobopod, Ecdysozoa

1. Summary

Tardigrades (water bears) are microscopic, segmented ecdysozoans with four pairs of legs. Lobopodous limbs that are similar to those seen in tardigrades are hypothesised to represent the ancestral state of Panarthropoda (Tardigrada + Onychophora + Arthropoda), and their evolutionary history is important to our understanding of ecdysozoan evolution. Equally important is our understanding of the functional morphology of these legs, which requires knowledge of their musculature. Tardigrade musculature is well documented but open questions remain. For example, while the muscular organisation of each trunk segment and its legs is unique, three of the four trunk segments are nevertheless relatively homonomous. To what extent, then, do leg muscles show segmental patterns? Specifically, which leg muscles are serially repeated and which are unique? The present study addresses these questions using a combination of techniques intended to visualise both the overall layout and fine structure of leg muscles. In doing so, we propose serial homologies for all leg muscles in each of the four legs and reveal new details of their cellular structure and attachment sites. We compare our results to those of previous studies and address the functional implications of specialised muscle cell morphologies.

2. Introduction

Tardigrades, commonly known as water bears, comprise a monophyletic group of microscopic ecdysozoans. While they are closely related to arthropods (moulting animals with jointed appendages), their external morphology most closely resembles that of the extant onychophorans (velvet worms) as well as several extinct lobopodians known from the fossil record [1, 2]. Tardigrades rely on short, unjointed, typically clawed appendages called lobopods for locomotion, which are remarkably similar in appearance to those of onychophorans despite differences in body size and habitat — onychophorans are exclusively terrestrial while tardigrades are aquatic or semiterrestrial. This resemblance, together with the hypothesis that lobopods represent the ancestral locomotory appendages of Panarthropoda [3, 4], fuel an incentive to study the functional morphology and, by extension, the musculature of the legs.

The muscular system of tardigrades has been well documented since the first detailed studies of tardigrade anatomy over a century ago [5-15]. Through careful examination of histological and whole-mount preparations, early researchers described the fine structure of tardigrade musculature in astonishing detail, even noting that a single muscle cell is composed of different regions [5-8]. Remarkably, many of these details were later confirmed when the ultrastructure of tardigrade muscles and their attachment sites was clarified using transmission electron microscopy [16-19]. The most recent investigations of myoanatomy have relied mainly on fluorescent labelling of filamentous actin — a major component of muscle fibres — in combination with confocal laser-scanning microscopy (CLSM) to generate high-resolution, three-dimensional datasets [11-15]. This in turn has allowed researchers to map the muscles that make up this organ system and compare them between tardigrade species. In doing so, clear patterns common to all tardigrade species have emerged. In all tardigrades examined to date, for example, the somatic musculature can be subdivided into dorsal, lateral, and ventral muscle groups (summarised in [20]). In the ventral group, muscles invariably attach at seven points arranged medially along the anterior-posterior body axis: two sites per trunk segment and one in the head. The dorsal and lateral longitudinal muscle strands span the length of the body from the head to the posteriormost body segment, while dorsoventral muscles bridge the dorsal and ventral attachment sites [14, 20].

Nonetheless, there is some disagreement between the myoanatomical descriptions of tardigrades in the literature, even between closely related species. For instance, although tardigrades are clearly segmented animals, there is uncertainty in the exact degree of metamerism, i.e. serial repetition along the anterior-posterior body axis, of the musculature [11, 14, 15, 21]. Segmentation hypotheses in many myoanatomical studies are based largely on the longitudinal and dorsoventral somatic musculature and the morphology of their attachment points [10, 14] with some support from the legs and their muscles, which are externally reflective of segmentation [21]. However, the low total number of body segments and the modified posteriormost leg pair complicates the assessment of serial homologies, and

*Author for correspondence (E-mail: vladimir.gross@uni-kassel.de).

consequently, the extent of segmentation of tardigrade musculature ranges in the literature from none or little [11, 22] to true of all muscle groups [14].

Another aspect that varies between descriptions is the musculature of the posteriormost leg pair. While all studies agree that the posteriormost leg is associated with the fewest muscles, the total number is debated [11, 15]. Whether these and other discrepancies are attributable to species differences or perhaps result from different techniques is unclear. In our view, this discussion may therefore benefit from a combined examination of both the organisation of the leg musculature as well as the fine structure of individual muscle cells, which to date has been done only for the marine eutardigrade *Halobiotus crispae* [12].

In order to understand the variation between different myoanatomical descriptions and to address the question of serial homologies, we reinvestigated the leg musculature of the model eutardigrade *Hypsibius exemplaris*. Using markers against both actin and myosin combined with super-resolution CLSM and 3D reconstruction, we map every muscle in each of the four legs and assess the serial homologies thereof. We also performed ultrastructural analysis in order to examine the fine structure of individual muscle cells and their attachment sites. We compare our results to those of previous investigations of the same and related tardigrade species and discuss the functional implications of specialised muscle cell morphologies.

3. Materials and Methods

Animals were maintained as described by Gross *et al.* [23]. In preparation for each experiment, specimens of *Hypsibius exemplaris* Gąsiorek *et al.*, 2018 were isolated from algae by rinsing with tap water through a 50- μ m nylon mesh and anaesthetised by asphyxiation in tap water at 60°C for 30 min.

3.1. Immunohistochemistry and DNA and F-actin labelling

The musculature of adult tardigrades was analysed using a combination of markers. For all experiments involving fluorescent markers, anaesthetised adults were immersed in 4% formalin buffered with PBS-Tx (0.1 mol L⁻¹ phosphate-buffered saline, pH 7.4, plus 1% Triton X-100) and fixed for 30 min–4 h at room temperature. During the fixation period, the cuticle of each specimen was punctured using electrolytically sharpened tungsten needles.

For fluorescent labelling of myosin, the samples were incubated in 50 mmol L⁻¹ NH₄Cl for 30 min following fixation in order to quench fixation-induced autofluorescence. Following several washes in PBS-Tx (2 × 10 min, 2 × 30 min, 2 × 1 h), the animals were incubated in a blocking solution containing 10% NGS (normal goat serum; Sigma-Aldrich, St. Louis, MO, USA) in PBS-Tx for 1 h at room temperature. The samples were then incubated in a solution containing mouse anti-myosin II monoclonal antibody (1:10 dilution) in PBS-Tx + 1% NGS + 0.02% NaN₃ overnight at room temperature. The monoclonal antibody (mouse anti-myosin II, DSHB Product 56-396-5), deposited by Günther Gerisch [25], was obtained from the Developmental Studies Hybridoma Bank (product 56-396-5), created by the NICHD of the NIH and maintained at The University of Iowa, Department of Biology, Iowa City, IA 52242. The specimens were then washed several times throughout the day in PBS-Tx, then incubated in a solution containing goat anti-mouse polyclonal antibody coupled to Alexa Fluor® 568 (1:500 dilution; Thermo Fischer Scientific, Waltham, MA, USA) in PBS-Tx + 1% NGS + 0.02% NaN₃ overnight at room temperature. Following several washes in PBS-Tx, the specimens were incubated in PBS containing 1 ng mL⁻¹ DAPI (4',6-diamidino-2-phenylindole; Thermo Fischer Scientific) for 15 minutes, rinsed quickly in PBS, and mounted between two glass coverslips in ProLong Gold (Invitrogen, Carlsbad, CA, USA). The coverslips were sealed the following day using nail polish.

For fluorescent labelling of filamentous actin (F-actin), fixed tardigrades were rinsed several times in PBS-Tx and incubated in a solution containing phalloidin-rhodamine (25 μ g mL⁻¹ in PBS; Thermo Fischer Scientific) overnight at 4°C. On the following day, the samples were rinsed with PBS and mounted as above. All fluorescently labelled specimens were imaged using either a Leica TCS STED (Leica Microsystems, Wetzlar, Germany) or a Zeiss LSM880 Airyscan (Carl Zeiss GmbH, Jena, Germany) confocal laser-scanning microscope.

3.2. Western blots

In order to confirm the specificity of the myosin II antibody, Western blots were performed according to the following protocol modified from Hering *et al.* [26] (Supplementary Figure 1). Several Petri dish cultures of *Hypsibius exemplaris* were concentrated by washing through a 50- μ m nylon mesh and collected in a 1.5 mL centrifuge tube. After adding up to 500 μ L of 5x Laemmli buffer and a protease inhibitor (cOmplete™, EDTA-free; Sigma-Aldrich), the tube was placed on ice and the tardigrades were manually homogenised using a plastic homogeniser. The contents were then heated for 10

min at 95°C, sonicated at 24 kHz using a UP200S Ultrasonic Processor (Hielscher Ultrasonics GmbH, Teltow, Germany), and heated again for 10 min. Proteins were separated using SDS-PAGE on a 7.5% gel for 45 min at 200 V. The proteins were transferred to a Porablot NCP nitrocellulose membrane (Macherey-Nagel GmbH & Co. KG, Düren, Germany) via wet blot and blocked with a solution containing 4% powdered milk in PBS for 30 min at room temperature. The samples were then incubated overnight with mouse anti-myosin II antibody (1:5 dilution), washed 6 × 5 min with PBS, incubated with goat anti-mouse antibody conjugated with alkaline phosphatase (1:10,000 dilution; dianova GmbH, Hamburg, Germany), and washed again with PBS. The signal was developed using a solution containing BCIP (50 µg/mL; Thermo Fischer Scientific) in dimethylformamide. The reaction was stopped by transferring to distilled water. In order to judge blotting efficiency, the original gel was subsequently stained with Brilliant Blue G 250 (Carl Roth GmbH, Karlsruhe, Germany).

3.3. Electron microscopy

For scanning electron microscopy (SEM), asphyxiated specimens were fixed in 4% formalin buffered with PBS overnight at 4°C. After several rinses in PBS, the specimens were transferred to a capsule with a pore size of 78 µm and dehydrated through an ascending ethanol series (10 min each at 30%, 50%, 70%, 90%, 2 × absolute ethanol). The capsule containing tardigrades was then transferred in absolute ethanol to a BAL-TEC CPD 030 critical point dryer (Balzers, Liechtenstein) and dried. Individual tardigrades were transferred to double-sided carbon tape on aluminium stubs using an eyebrow hair. In order to expose the inside of the tardigrade, one stub containing several specimens was pressed against a second, empty stub with carbon tape and the two were pulled apart, splitting the specimens. All samples were then sputter coated with ~30 nm of gold-palladium using a Polaron SC7640 sputter coater (Quorum Technologies, Kent, UK) and analysed using a Hitachi S-4000 field emission scanning electron microscope (Hitachi High-Technologies Europe GmbH, Krefeld, Germany) at an accelerating voltage of 5 kV.

Ultrastructural analysis was performed via scanning transmission electron microscopy (STEM). For this purpose, asphyxiated specimens were fixed in 4% formalin + 1% glutaraldehyde in 0.067 mol L⁻¹ Sørensen phosphate buffer [27] for 1 h at room temperature. After 2 × 15 min washes in buffer, the samples were post-fixed in 1% OsO₄ in Sørensen buffer for 1 h at room temperature. The samples were then washed several times quickly and then overnight in distilled water. On the following day, the samples were dehydrated through an increasing ethanol series as above followed by 2 × 10 min in acetone. The tardigrades were then embedded in epoxy resin using a Spurr Low-Viscosity Embedding Kit (Sigma-Aldrich) as follows: Specimens were first incubated in a 1:1 mixture of acetone and Spurr resin (using the manufacturer's "standard" formulation: ~~4.1 g ERL 4221, 1.43 g DER 736, 5.9 g NSA, 0.1 g DMAE~~) for 30 min at room temperature, followed by pure Spurr resin for 2 × 1 h and then overnight. On the following day, the specimens were transferred to fresh Spurr resin and poured into plastic box-shaped moulds. The specimens were oriented individually within the moulds using fine needles and polymerised at 70°C for 3 days. Sectioning was done with glass knives using a Reichert-Jung Ultracut E ultramicrotome (C. Reichert AG, Vienna, Austria). Serial sections displaying silver-gold interference colours were collected on single-slot copper grids coated with Formvar[®]/Vinylec[®] (Plano GmbH, Wetzlar, Germany) and contrasted for 5 min with a uranyl acetate replacement stain (UAR-EMS, Electron Microscopy Sciences, Hatfield, PA, USA; 1:4 dilution in 50% methanol [28]) and for 8 min with lead citrate [29]. Samples were imaged on a custom-made STEM-in-SEM grid holder using a Hitachi S-4000 field emission scanning electron microscope at an acceleration voltage of 30 kV. Images of serial sections were aligned using the TrakEM2 plugin for FIJI [30-32].

3.4. Image processing and segmentation

Raw Airyscan image stacks were processed in Zen Blue (Carl Zeiss) with filter strength set to "auto". CLSM substacks were assembled and adjusted for brightness and contrast using the FIJI distribution of ImageJ2 [33]. The SEM images were adjusted for brightness and contrast and artificially coloured using Adobe Photoshop CS6 (Adobe Systems, San Jose, CA, USA). Specimens that showed an even, high-quality fluorescent signal were chosen for further segmentation, which was done manually using Amira 5.4.0 (Thermo Fischer Scientific). For creating 3D representations, surfaces of the individually segmented muscles were superimposed over a volume-rendering of the original CLSM stack. For creating CLSM substacks featuring individual artificially coloured muscles, the exported label fields of segmented muscles were used as a mask over the original CLSM data. All 3D images and CLSM substacks shown are of only one side of the body. Supplementary movies were created using Amira 5.4.0 and assembled using Adobe Premiere Pro CS6 (Adobe Systems). All diagrams, labelling, and assembly of image plates were done using Adobe Illustrator CS6 (Adobe Systems).

4. Results

4.1. Nomenclature and morphology of muscles and their attachment sites

Fluorescent labelling of F-actin revealed all muscles in the tardigrade body. Our analysis is focused on the leg musculature, which accounts for a large proportion of the total body musculature (figures 1*a–d*, 2, 3, 4*a–e*, 5*a–d*, Supplementary Movies 1 and 2). Combined fluorescent labelling of myosin II, F-actin, and DNA show that some muscles contain two or more contractile strands but only a single nucleus (figures 6*a–d* and 7*a–d*). In these cases where a muscle cell branches, each strand has its own proximal attachment site, but all strands merge distally. This construction is confirmed by scanning electron microscopy of bisected specimens (figure 6*a*). In this work, we therefore define an individual muscle based on the presence of one nucleus and consider several strands that share a nucleus as a single muscle.

Since a single muscle may have several attachment sites and a single attachment site may be shared by several muscles, we created a nomenclature that is independent from attachment sites in order to avoid ambiguity (see Supplementary Table 1 for a comparison with previous nomenclatures). In naming leg muscles, we use specific labels for hypothesised serially homologous muscles (Table 1). Each muscle is designated by the leg it supplies (i, ii, iii, or iv) followed by its specific number (e.g., iM2 for muscle M2 of the first leg, iiM2 for muscle M2 of the second leg, etc.) (figure 2). In the following description, we refer to a specific muscle (e.g., iiM2) or, alternatively, to all serial homologues of a muscle by not designating a leg (e.g., M2). Hypotheses of serial homologies are based on i) the position of all attachment sites of each muscle, both distal (closest to the tip of the leg) and proximal (within or closest to the trunk), ii) the spatial relationships between muscles and muscle groups and iii) cellular morphology of individual muscles, where applicable. Muscles were named in order from anterior to posterior and dorsal to ventral (in that order) according to their proximal attachment sites, except for muscles of leg IV, which were named according to their serial homologues in legs I–III (figures 2, 3, and Table 1).

The distal attachment site of each leg muscle can generally be assigned to either the anterior or posterior region of each leg, where muscle attachment sites are concentrated (figure 5*a–d*). Muscle attachment sites exhibit a distinct morphology that allows them to be recognized both externally and internally (figures 8*a–d* and 9). Scanning electron microscopy reveals that externally, areas associated with muscle attachment sites exhibit what appear to be a series of tiny holes and, in some cases, a ridge or fold in the cuticle (figures 8*a–d* and 9). Ultrastructural analysis shows that the holes do not form complete channels but rather correspond to invaginations of the cuticle that are associated with filaments from the muscle attachments (figure 10*a–d*). The ridges, on the other hand, correspond to thickenings of the cuticle, specifically the electron-lucent procuticular layer (*sensu* Shaw [16]; figure 10*a–d*). Otherwise, our data are largely in line with previous studies, which show a relatively invariant ultrastructure of somatic muscle attachments among all tardigrade species examined [16, 18, 19].

4.2. Musculature of the legs

In each leg, the muscles are arranged around the periphery of the leg — generally either in the anterior or posterior region of the leg — while the large space in the middle is filled with haemolymph (figure 4*a–e*, 5*a–d*, 8*a–d*, and Supplementary Movie 2). The anterior region contains a higher number of muscles than the posterior region (figures 4*e*). Our data confirm that the number of muscles per leg decreases from anterior to posterior, i.e. that leg I has the highest number of leg muscles and leg IV has the lowest number (table 1). Specifically, we identified 14, 12, 11, and 10 muscles in legs I, II, III, and IV, respectively (table 1).

Two muscles are unique to leg I (iM1 and iM10) and do not share a serial homologue in any other leg (figures 3, 4*a*, and table 1). Both belong to the group of leg muscles located in the anterior region of the leg. Muscle iM1 is the anteriormost leg muscle in the body, running laterally next to the pharynx (figure 4*a*). Muscle iM10 runs parallel to iM9, both of which are very short and attach proximally at ventral attachment site 1 (*sensu* Smith and Jockusch [15]), with iM9 attaching on the contralateral side of this attachment site (figure 4*a, e*, and movie S1). Both iM9 and iM10 run very close to and share the ventral attachment site with iM11, which attaches distally at the anterior tip of the leg (figure 4*a, e*). In legs II–IV, the proximal side of M9 crosses into the anterior adjacent segment and attaches at the ventral attachment site there, while M11 stays in the same segment (figures 3 and 4*b–d*).

The M2 muscles attach distally at the lateral base of the leg and proximally at the dorsal longitudinal musculature (figure 3). In legs II–IV, they represent the anteriormost dorsally attaching leg muscles; iiiM2 is unique in that it shows a branched morphology (figures 3, 4*b–d*, and 6*b*).

Two additional muscle groups, M3 and M4, have distal attachments at approximately the same level as M2, i.e. at approximately half the length of the leg, and both attach on the lateral side (figures 3 and 4*a–d*). Muscles M3 and M4 share a proximal attachment site located at an intermediate position of the body wall between the lateral and dorsal longitudinal muscle groups (figures 3 and 4*a–d*). The serial homologue of M3 is missing in leg IV (figure 3).

The M5 muscles attach distally at the anterior tip of each leg and branch proximally, with all pairs of proximal attachment sites located along the dorsal longitudinal musculature in legs I–III (figures 3 and 4*a–d*). The nucleus of each M3 muscle is located between the two strands, shortly above where the strands split (figure 6*b*). Muscle ivM5 instead splits into three strands proximally, with the nucleus located further distally, before the three strands split (figure 6*c, d*). The two posteriormost proximal attachment sites of ivM5 are shared with those of ivM6 (figure 4*d*).

The M6 muscles branch into two strands in legs I, II, and IV, where they attach distally at the posterior base of each leg and proximally along the dorsal longitudinal musculature (figures 3, 4*a–d*, and 7*a, b*). Muscle iiiM6 is different in that it splits into three branches, with the posteriormost branch attaching along the lateral body wall (figures 3, 4*c*, and 7*c, d*). In addition, the distal attachment site of iiiM6 is at approximately half the length of the leg, although this muscle also retains an intermediate attachment site at the base of the leg as in legs I and II (figure 4*c*). The posteriormost strand of iiiM6 corresponds in attachment sites and course to iM7 and iiM7, thereby superficially appearing to be the serial homologue of M7 (figure 7*a–d*). However, myosin II and nuclear labelling reveal that this strand together with the other two strands of iiiM6 in fact constitute a single cell (figure 7*c–d*). In all M6 muscles, the nucleus is located after the split and between the strands, specifically between the two posterior strands in iiiM6 (figure 7*a–d*).

The M7 muscles represent the posteriormost-reaching muscles of legs I and II (figures 2, 3, and 4*a, b*). These muscles are linear in morphology but show three attachment sites: one proximal along the lateral body wall, one distal at approximately half the length of the leg, and one intermediate site at the posterior base of the leg (figures 4*a, b*, 10*a–d*, and 11*a, b*). Each intermediate site of M7 is shared with the distal attachment site of M6, where M7 attaches both to the wall of the leg as well as directly to M6 (figure 10*a–d*). Leg IV also lacks a serial homologue of M7 (figure 3).

The distalmost posterior region of each leg is largely occupied by an M14 muscle; a wide, flattened muscle that follows the curve of the posterior margin of the leg (figures 3, 4*a–e*, 11*a* and Supplementary Movie 3). In each M14 muscle, the nucleus and neuromuscular junction are both located near the proximal attachment site (figure 11*a–c* and Supplementary Movie 3), which is shared with three additional muscles: M7, M8, and M12 in legs I and II or M6, M8 and M12 in legs III and IV (figure 4*a–d*). Together, this attachment site forms a wide crease on the posterior cuticle of each leg (figure 8*b, c*). The M8 muscles, which represent the lateral muscles of these groups of three, attach along the lateral body wall while M12 — the medial muscles — attach at ventral attachment sites (figures 3 and 4*a–d*). In leg IV, muscle ivM12 instead represents the middle muscle of the group, with ivM6 being medial (figure 4*d*). Finally, the distal attachment site of each M14 muscle is shared with an M13 muscle, which are short and thin muscles that run anteriorly within the leg to an attachment site shared with an M9 muscle (figures 3 and 4*a–d*). The M13 and M14 muscles represent true intrinsic muscles, in that they are located in their entirety within the leg.

4.3. Remarks on Leg IV

The musculature of leg IV is modified considerably compared to that of the anterior three leg pairs, reflecting its derived external morphology. We took the evolutionary rotation of leg IV into account when analysing potential serial homologies. Compared to the other three leg pairs, muscle M14 is more elongate and occupies an anteroventral rather than posterior position within the leg (figures 3 and 4*d*). Muscles ivM8 and ivM9 are longer than their serial homologues in legs I–III, reflecting the elongate nature of the posterior trunk and posteriormost legs (figures 3 and 4*d*). The remaining muscles, although superficially quite different from those of the other legs, follow the same relative paths and attach at corresponding attachment sites in the leg and body.

5. Discussion

5.1. Fine structure and functional implications

Muscle cells with several contractile strands presumably function as a single unit, whereby the contraction of all strands occurs simultaneously. This may be a mechanism to reduce the strain on any single attachment site by distributing the generated force over two or more sites. In onychophorans, for example, some leg muscles are organised as a network of thin fibres and spread out over a relatively large surface in order to minimise the physical deformation of the leg during muscle contraction [34]. The same principle may apply to branched muscles in the tardigrade leg, albeit on the scale of only two or three fibres. A similar branching of individual muscle cells is also known from other metazoans, such as in the caudal spine of nematomorph larvae [35] and the pharynx of a ctenophore [36]. Based on the positions of muscles exhibiting specialised morphologies — which we define as any shape other than the linear strap-shape of somatic muscles — and their presumptive roles in locomotion, we hypothesise that derived muscle morphologies in tardigrade legs are the result of functional requirements above all [37], rather than e.g. evolutionary constraints [38]. This hypothesis is in line with the conclusions of Walz [37] regarding somatic and visceral muscle types in tardigrades, and of Heumann & Zebe [39] regarding the distribution of obliquely striated muscles across annelids, molluscs, and nematodes. On the other hand, the morphology of attachment sites and the layout of the muscular system has been shown to be phylogenetically informative within Tardigrada, at least at higher taxonomic levels [14].

Our hypothesis regarding functional constraints does not only apply to the branched muscles. The position of iM1 as the anteriormost muscle in leg I, for example, may explain why leg I can be extended much further forward than any other leg [40], considering that there are no serial homologues of M1 in legs II–IV. At the posterior of the leg, the morphology of the M14 muscles (the "sheet muscles" *sensu* Smith and Jockusch [15]) implicates their role in coordinated contraction of the leg or retraction of the claws, either during locomotion or tun formation. The shape of the M14 attachment sites, i.e. wide horizontal creases along the posterior of legs I–III may serve to better control the plane in which each leg folds and prevent its collapse toward a single small point during muscle contraction. The anteroventral position of ivM14 may also facilitate the grasping action that has been proposed to be one of the main functions of leg IV [40]. Similarly, the M7 muscles may be responsible for the retraction of the legs, and the presence of an intermediate attachment site along the length of each of these muscles may prevent kinks in the base of the leg during contraction. It should be noted that in leg III, the M7 homologue appears to have been replaced by the posteriormost strand of the triple-branched iiiM6 both in form and, presumably, in function. Alternatively, an ancestral branched muscle may have split into separate cells in legs I and II for more precise control of individual muscles. We favour the first scenario, as a higher number of leg muscles is seen in the other extant lobopodians, i.e., the onychophorans [34], and a reduction might have occurred in tardigrades at several levels due to miniaturization [41]. Tardigrade leg musculature is reduced not only in the number of muscles, but also in complexity compared to that of onychophorans. For example, although most muscles are arranged around the periphery of the leg in both taxa, onychophorans possess additional leg muscles that partition the leg into compartments, as well as twisted muscles that are responsible for the rotation of the leg [34]. The lack of the latter in tardigrades partially explains the simple back-and-forth movement of the legs (without rotation) characteristic of tardigrade locomotion [40]. Whatever the case may be, we hypothesise that a reduced number but higher degree of branching of muscles towards the posterior of the animal (e.g., iiiM2, iiiM6, and ivM5) may be related to the observation that movement of the posterior legs is not as precise during locomotion as that of the anterior two leg pairs [40]. A comparison of leg musculature between tardigrade species with different gaits is necessary to test this hypothesis.

5.2. Comparison of myoanatomy with previous studies

Several previous studies have mapped out the muscular system of different tardigrade species. Investigations of the musculature of *Hypsibius exemplaris* or closely related species [10, 11, 14, 15] generally agree on the layout of attachment sites with little deviation (see supplementary material, table S1 for a comparison). While our data are largely in line with previous studies in terms of the positions of muscle attachment sites, the leg muscles themselves are more difficult to compare. This challenge partly comes from the fact that the nomenclature traditionally used in the literature is mainly focused on attachment sites and does not always account for the fact that branched leg muscles have more than two attachment sites. Müller [10] was able to identify most of the branched leg muscles but named each strand according to its attachment sites regardless. Her depiction of the sheet muscle (iiM14 herein) and the arrangement of muscles associated with its proximal attachment site (iiM7, iiM8, and iiM12 herein) largely mirror our results (compare with fig. 4B in [10]). The exception is that our data show that the split between muscles "11" and "i" illustrated by Müller [10] actually corresponds to the distal attachment site of iiM6 and the intermediate attachment site of iiM7, respectively. In this case, this disparity might be attributable to species-specific differences, as the illustration of Müller [10] is based on *Macrobiotus hufelandi*. In fact, Müller [10] notes that even the closely related *Hypsibius convergens* and *Ramazzottius oberhaeseri* show differences in the number and grouping of appendage muscles.

More recent studies based on fluorescent labelling of F-actin in *Hypsibius exemplaris* [15] and *Hypsibius* sp. [11] generally differ only in details of individual muscles. Although the musculature of leg IV is the greatest point of disagreement, previous studies still consistently show that leg IV contains the fewest muscles, with a maximum total of seven [10, 11, 15]. This notable deviation from the ten muscles we describe herein may be due to a number of factors. It is possible that the super-resolution CLSM module used for the present study allowed us to identify muscles or parts of muscles that were previously overlooked using conventional techniques, for example muscle ivM13. However, ivM13, the smallest and thinnest muscle in leg IV, was already described by Müller [10] in *Ramazzottius oberhaeseri* (muscle "P₁-an") and is also visible in preparations of other species, including *Hypsibius* sp. (cf. fig. 3a in [11]), *Paramacrobiotus richtersi*, *Milnesium* cf. *tardigradum*, *Acutuncus antarcticus* (cf. figs 6, 7 in [14]), and the previous investigation of *Hypsibius exemplaris* (cf. fig. 7C in [15]). Therefore, while the improved resolution might uncover fine details that may help to inform serial homologies, it did not reveal any muscles that were previously undiscovered. Ironically, the non-contractile region of a branched muscle cell that spans the two contractile strands is invisible to phalloidin labelling, which may partly explain why our combined approach using anti-myosin II immunolabelling revealed details that in some cases more closely match earlier studies based on bright-field observations (e.g. [10]) rather than more recent F-actin-based investigations (e.g. [11, 15]).

We find it more plausible that at least some differences can be attributed to a matter of interpretation, given that our description of the sheet muscles (M14) and muscles associated with their proximal attachment sites more closely matches that of the more distantly related *Halobiotus crispae* [12] than previous investigations of *Hypsibius exemplaris* [15] or *Hypsibius* sp. [11]. Specifically regarding leg IV, we believe that the major difference in the number of described muscles between the present and previous studies can be ascribed to the unique morphology of the leg itself. Leg IV faces backwards and essentially appears as a posterior extension of the trunk, complicating the distinction between the leg itself and the posteriormost trunk segment. This modification relative to legs I–III is especially evident in the sheet muscle (ivM14), which is not only longer than its serial homologs in legs I–III, but is also aligned more with the anterior–posterior body axis rather than the dorsoventral body axis as in the other legs. If leg IV does indeed possess rotated serial homologues of most of the muscles found in legs I–III and this pattern is common to other tardigrade species, it could explain why its movement in *Echiniscus testudo* and *Macrobiotus* sp. was described as similar to but opposite of legs I–III ("ähnlich, aber entgegengesetzt," [40]). Taking these modifications into account and viewing the muscles in this light lead us to the conclusion that the musculature of leg IV is, though slightly reduced, less derived than previously thought [14].

6. Conclusion

The results presented herein show that virtually all leg muscles are serially repeated, adding further support for the hypothesis that tardigrade musculature is largely a metamerically arranged organ system [14, 15, 21]. The striking reduction in the total number of leg muscles from anterior to posterior combined with an increased degree of branching begs the question of how such an arrangement evolved. It is known that in *H. exemplaris*, the segmental trunk ganglia arise from anterior to posterior during development [42]. It is possible that the musculature follows the same sequence, in which case heterochronic shifts in development might be responsible for the observed differences in musculature between the different legs. Along these lines, is increased branching of muscle strands a functional compensation for the reduction in the number of muscle cells, or perhaps *vice versa*? Studies of embryonic and postembryonic development may help address these questions by documenting how and when each muscle arises, particularly regarding the branched muscles. It has been shown previously that newly hatched juveniles lack the tiny cuticular pores indicative of muscle attachment sites [43], suggesting that the cuticle or musculature might continue developing to some extent postembryonically. Do the underlying muscles in juveniles also differ morphologically from those in adults? Finally, comparative studies of the terminal or otherwise highly derived segments of onychophorans or fossil lobopodians may provide further insights into the nature of serial homologies in these animals (e.g., [44]).

Acknowledgments

Sonja Fuhrmann and Sonja Kasten (University of Kassel) are acknowledged for assistance with Western blotting. The authors thank Ivo de Sena Oliveira (University of Kassel) and Sarah Atherton (Swedish Museum of Natural History) for constructive comments and critical reading of the manuscript. We thank Markus Maniak and Heike Otto (University of Kassel) for preparing and providing the hybridoma supernatant.

Ethical Statement

The experiments in this study did not require an approval by an ethical committee. All procedures in this investigation complied with international and institutional guidelines, including the guidelines for animal welfare as laid down by the German Research Foundation (DFG).

Data Accessibility

The datasets supporting the conclusions of this article are included within the article and its additional files.

Competing Interests

The authors declare no competing interests.

Authors' Contributions

Conception and design: VG and GM. Acquisition of data: VG. Analysis and interpretation of data: VG and GM. Drafting of manuscript: VG. Revision of drafts: VG and GM. Final approval of submitted version: VG and GM.

References

1. Ou Q, Shu D, Mayer G. 2012 Cambrian lobopodians and extant onychophorans provide new insights into early cephalization in Panarthropoda. *Nat. Commun.* **3**, 1261. (doi:10.1038/ncomms2272)
2. Smith MR, Ortega-Hernández J. 2014 *Hallucigenia*'s onychophoran-like claws and

- the case for Tactopoda. *Nature*. **514**, 363–366. (doi:10.1038/nature13576)
3. Liu J, Dunlop JA. 2014 Cambrian lobopodians: A review of recent progress in our understanding of their morphology and evolution. *Palaeogeogr. Palaeoclimatol. Palaeoecol.* **398**, 4–15. (doi:10.1016/j.palaeo.2013.06.008)
4. Ortega-Hernandez J. 2015 Lobopodians. *Curr. Biol.* **25**, R845–R875. (doi:10.1016/j.cub.2015.07.028)
5. Doyère M. 1840 Mémoire sur les Tardigrades. *Ann. Sci. Nat. (Paris), Zool., Ser. 2*. **14**, 269–361.
6. Greeff R. 1865 Über das Nervensystem der Bärthierchen, Arctiscoida C.A.S. Schultze (Tardigraden Doyère) mit besonderer Berücksichtigung der Muskelnerven und deren Endigungen. *Arch. Mikrosk. Anat.* **1**, 101–122.
7. Plate L. 1889 Beiträge zur Naturgeschichte der Tardigraden. *Zool. Jahrb. Abt. Anat. Ontog. Tiere.* **3**, 487–550.
8. Basse A. 1906 Beiträge zur Kenntnis des Baues der Tardigraden. *Z. wiss. Zool.* **30**, 259–281.
9. Baumann H. 1921 Beitrag zur Kenntnis der Anatomie der Tardigraden (*Macrobotus hufelandi*). *Z. wiss. Zool.* **118**, 637–652.
10. Müller J. 1936 Zur vergleichenden Myologie der Tardigraden. *Z. wiss. Zool.* **147**, 171–204.
11. Schmidt-Rhaesa A, Kulesa J. 2007 Muscular architecture of *Milnesium tardigradum* and *Hypsibius* sp. (Eutardigrada, Tardigrada) with some data on *Ramazottius oberhaeuseri*. *Zoomorphology*. **126**, 265–281. (doi:10.1007/s00435-007-0046-0)
12. Halberg KA, Persson D, Møbjerg N, Wanninger A, Kristensen RM. 2009 Myoanatomy of the marine tardigrade *Halobiotus crispae* (Eutardigrada: Hypsibiidae). *J. Morphol.* **270**, 996–1013. (doi:10.1002/jmor.10734)
13. Schulze C, Schmidt-Rhaesa A. 2011 Organisation of the musculature of *Batillipes pennaki* (Arthrotardigrada, Tardigrada). *Meiofauna Mar.* **19**, 195–207.
14. Marchioro T, Rebecchi L, Cesari M, Hansen JG, Viotti G, Guidetti R. 2013 Somatic musculature of Tardigrada: phylogenetic signal and metameric patterns. *Zool. J. Linn. Soc.* **169**, 580–603. (doi:10.1111/zoj.12079)
15. Smith FW, Jockusch EL. 2014 The metameric pattern of *Hypsibius dujardini* (Eutardigrada) and its relationship to that of other panarthropods. *Front. Zool.* **11**, 66. (doi:10.1186/s12983-014-0066-9)
16. Shaw K. 1974 The fine structure of muscle cells and their attachments in the tardigrade *Macrobotus hufelandi*. *Tissue Cell.* **6**, 431–445.
17. Walz B. 1974 The fine structure of somatic muscles of Tardigrada. *Cell Tissue Res.* **149**, 81–89.
18. Greven H, Grohé G. 1975 Die Feinstruktur des Integumentes und der Muskelansatzstellen von *Echiniscoides sigismundi* (Heterotardigrada). *Helgol. Mar. Res.* **27**, 450–460. (doi:10.1007/BF01611149)
19. Kristensen RM. 1978 On the structure of *Batillipes noerrevangi* Kristensen, 1978. 2. The muscle-attachments and the true cross-striated muscles. *Zool. Anz.* **200**, 173–184.
20. Møbjerg N, Jørgensen A, Kristensen RM, Neves RC. 2018 Morphology and Functional Anatomy. In *Water Bears: The Biology of Tardigrades* (ed. RO Schill), pp. 57–94. Cham, Switzerland: Springer Nature Switzerland.
21. Smith FW, Goldstein B. 2017 Segmentation in Tardigrada and diversification of segmental patterns in Panarthropoda. *Arthropod Struct. Dev.* **46**, 328–340. (doi:http://dx.doi.org/10.1016/j.asd.2016.10.005)
22. Edgecombe GD. 2010 Arthropod phylogeny: An overview from the perspectives of morphology, molecular data and the fossil record. *Arthropod Struct. Dev.* **39**, 74–87.
23. Gross V, Bährle R, Mayer G. 2018 Detection of cell proliferation in adults of the water bear *Hypsibius dujardini* (Tardigrada) via incorporation of a thymidine analog. *Tissue Cell.* **51**, 77–83. (doi:10.1016/j.tice.2018.03.005)
24. Gąsiorek P, Stec D, Morek W, Michalczyk L. 2018 An integrative redescription of *Hypsibius dujardini* (Doyère, 1840), the nominal taxon for Hypsibiodea (Tardigrada: Eutardigrada). *Zootaxa.* **4415**, 45–75. (doi:10.11646/zootaxa.4415.1.2)
25. Pagh K, Gerisch, G. 1986 Monoclonal antibodies binding to the tail of *Dictyostelium discoideum* myosin: their effects on antiparallel and parallel assembly and actin-activated ATPase activity. *J. Cell Biol.* **103**, 1527–1538. (doi:10.1083/jcb.103.4.1527)
26. Hering L, Bouameur J-E, Reichelt J, Magin TM, Mayer G. 2016 Novel origin of lamin-derived cytoplasmic intermediate filaments in tardigrades. *eLife.* **5**, e11117. (doi:10.7554/eLife.11117)
27. Sørensen SPL. 1909 Über die Messung und die Bedeutung der Wasserstoffionenkonzentration bei enzymatischen Prozessen. *Biochem. Z.* **21**, 131–200.
28. Nakakoshi M, Nishioka H, Katayama E. 2011 New versatile staining reagents for biological transmission electron microscopy that substitute for uranyl acetate. *J. Electron Microsc.* **60**, 401–407. (doi:10.1093/jmicro/dfr084)
29. Reynolds ES. 1963 The use of lead citrate at high pH as an electron-opaque stain in electron microscopy. *J. Cell Biol.* **17**, 208–212.
30. Saalfeld S, Cardona A, Hartenstein V, Tomančák P. 2010 As-rigid-as-possible mosaicking and serial section registration of large ssTEM datasets. *Bioinformatics.* **26**, i57–i63. (doi:10.1093/bioinformatics/btq219)
31. Saalfeld S, Fetter R, Cardona A, Tomančák P. 2012 Elastic volume reconstruction from series of ultra-thin microscopy sections. *Nat. Meth.* **9**, 717. (doi:10.1038/nmeth.2072)
32. Cardona A, Saalfeld S, Schindelin J, Arganda-Carreras I, Preibisch S, Longair M, Tomančák P, Hartenstein V, Douglas RJ. 2012 TrakEM2 software for neural circuit reconstruction. *PLoS ONE.* **7**, e38011. (doi:10.1371/journal.pone.0038011)
33. Rueden CT, Schindelin J, Hiner MC, DeZonia BE, Walter AE, Arena ET, Eliceiri KW. 2017 ImageJ2: ImageJ for the next generation of scientific image data. *BMC Bioinformatics.* **18**, 529. (doi:10.1186/s12859-017-1934-z)
34. Oliveira IS, Kumerics A, Jahn H, Müller M, Pfeiffer F, Mayer G. (in press) Functional morphology of a lobopod: case study of an onychophoran leg. *R. Soc. open sci.* (this issue)
35. Müller MCM, Jochmann R, Schmidt-Rhaesa A. 2004 The musculature of horsehair worm larvae (*Gordius aquaticus*, *Paragordius varius*, Nematomorpha): F-actin staining and reconstruction by cLSM and TEM. *Zoomorphology.* **123**, 45–54. (doi:10.1007/s00435-003-0088-x)
36. Norekian TP, Moroz LL. 2018 Neuromuscular organization of the ctenophore *Pleurobrachia bachei*. *J. Comp. Neurol.* **527**, 406–436. (doi:10.1002/cne.24546)
37. Walz B. 1975 Ultrastructure of muscle cells in *Macrobotus hufelandi*. *Mem. Ist. Ital. Idrobiol.* **32**, 425–443.
38. Budd GE. 2001 Tardigrades as 'stem-group arthropods': The evidence from the Cambrian fauna. *Zool. Anz.* **240**, 265–279.
39. Heumann H-G, Zebe E. 1967 Über Feinbau und Funktionsweise der Fasern aus dem Hautmuskelschlauch des Regenwurms, *Lumbricus terrestris* L. *Z. Zellforsch.* **78**, 131–150. (doi:10.1007/BF00344407)
40. Schüttler L, Greven H. 2000/2001 Beobachtungen zur Lokomotion von Tardigraden. *Acta Biol. Benrodis.* **11**, 31–50.
41. Gross V, Treffkorn S, Reichelt J, Epple L, Lüter C, Mayer G. 2019 Miniaturization of tardigrades (water bears): morphological and genomic perspectives. *Arthropod Struct. Dev.* **48**, 12–19. (doi:10.1016/j.asd.2018.11.006)
42. Gross V, Mayer G. 2015 Neural development in the tardigrade *Hypsibius dujardini* based on anti-acetylated α -tubulin immunolabeling. *EvoDevo.* **6**, 12. (doi:10.1186/s13227-015-0008-4)
43. Gross V, Minich I, Mayer G. 2017 External morphogenesis of the tardigrade *Hypsibius dujardini* as revealed by scanning electron microscopy. *J. Morphol.* **278**, 563–573. (doi:10.1002/jmor.20654)
44. Oliveira IS, Mayer G. 2013 Apodemes associated with limbs support serial homology of claws and jaws in Onychophora (velvet worms). *J. Morphol.* **274**, 1180–1190. (doi:10.1002/jmor.20171)

Figure and table captions

Figure 1. Overview of the muscular system of the eutardigrade *Hypsibius exemplaris*. Colour-coding according to leg. Lateral view; anterior is left, dorsal is up in both images. (a) CLSM substack showing the muscular system of the left half of the animal, with leg muscles artificially coloured by leg. F-actin labelling. (b) 3D reconstruction based on the CLSM dataset shown in (a). Abbreviations: he, head; I–IV, legs I–IV. Scale bar: 20 μm (in both images).

Figure 2. Leg muscles of the tardigrade *Hypsibius exemplaris*. Colour-coding according to hypothesised serial homologies. 3D reconstruction based on CLSM data. Lateral view; anterior is left, dorsal is up. Scale bar: 20 μm .

Figure 3. Hypothesised serially homologous leg muscles, individually highlighted. 3D reconstruction based on CLSM data. Lateral view; anterior is left, dorsal is up. Scale bar: 20 μm (for all images).

Figure 4. Leg muscles of the tardigrade *Hypsibius exemplaris*. 3D reconstructions of legs I–IV (a–d, respectively) based on CLSM datasets; F-actin labelling. Dotted lines indicate planes of cross-sections shown in (e). Colour-coding according to hypothesised serial homologies. Lateral view; anterior is left, dorsal is up. See Fig. 1 for muscle names. (a) Muscle iM1 (red) does not share a serial homolog in any other leg. (c) Muscle iiiM2 (green) shows a branching morphology, unique to leg III. Muscle iiiM6 (blue) consists of three strands, with the posteriormost strand possibly representing a functional analogue of iM7 and iiM7. (d) Leg IV is rotated backwards so that ivM14 occupies an anteroventral position within the leg. Muscle ivM5 (pink) consists of three strands. (e) Schematic diagrams illustrating cross-sections of legs I–IV. Virtual sectioning planes are indicated by dotted lines in (a–d); top row corresponds to upper plane, bottom row to lower plane. Notice how muscles of legs I–III are concentrated either at the anterior or posterior of the leg, with muscle M13 spanning the two regions. Muscle M14 occupies the majority of the distal posterior of legs I–III. Cross-sections not drawn to scale. Orientation in (e) is for legs I–III. Orientation legends: d, dorsal; m, medial; p, posterior. Scale bars: 10 μm (a–d).

Figure 5. Leg muscles of the tardigrade *Hypsibius exemplaris* from the distal perspective. Colour-coding according to hypothesised serial homologies. 3D reconstructions of legs I–IV (a–d, respectively) based on CLSM datasets; F-actin labelling. Notice how muscles are arranged around the periphery of the leg — primarily in the anterior and posterior regions of each leg — with a large space in the middle (asterisk). The M13 muscles are the only muscles that cross between the anterior and posterior regions. (d) The muscles of leg IV are laterally compressed, presenting a smaller empty space between the muscles. Orientation in (c) for (a–c). Orientation legends: d, dorsal; m, medial; p, posterior.

Figure 6. Branched morphologies in leg muscles. Lateral views; anterior is left, dorsal is up in all images. (a) False-coloured scanning electron micrograph of a bisected specimen showing iM5. (b, c) CLSM substacks; anti-myosin II immunolabelling (glow) and nuclear counterstain (cyan). (b) Leg III. The branched muscles iiiM2 and iiiM5 with their respective nuclei (arrowheads) located between where the two strands of each muscle split. (c, d) Muscle ivM5 and its three strands. Notice how its nucleus (arrowhead) is located below the point where the three strands split. Scale bars: 3 μm (a); 10 μm (b, c).

Figure 7. Legs II and III showing the relationship between muscles M6 and M7. Lateral views of CLSM substacks; anterior is left, dorsal is up. Anti-myosin II immunolabelling (glow) and nuclear counterstain (cyan). (a, b) Muscles iiM6 and iiM7, each with their own nucleus (arrowheads). (c, d) Muscle iiiM6, showing three strands and one nucleus (arrowhead). Scale bars: 5 μm (a–d).

Figure 8. Lateral and posterior leg muscle attachment sites seen on the external cuticle. Scanning electron micrographs. (a) Lateral and distal anterior muscle attachment sites of leg III. Each attachment site is recognisable by what appear to be tiny holes in the cuticle (arrows) and, in some cases, a ridge or fold. Lateral view; anterior is left, dorsal is up. (b–d) Attachment sites of iM14. Posterior view of leg I; medial is right, dorsal is up. Notice the wide, thick folds of the cuticle at both the proximal (c) and distal (d) sheet muscle attachment sites. Scale bars: 5 μm (a); 500 nm (insets in a); 3 μm (b); 1 μm (c, d).

Figure 9. Posterior end of the body showing the attachment sites associated with leg IV. Scanning electron micrographs. Posterodorsal view; anterior is upper left. Muscle attachment sites are recognisable by what appear to be tiny holes in the cuticle. Abbreviation: cm, attachment site of the cloacal muscles. Scale bars: 4 μm (overview); 500 nm (close-ups of a–c); 2 μm (close-up of d).

Figure 10. Fine structure of muscle M7. STEM-in-SEM images. Sagittal sections; anterior is left, dorsal is up. (a) Overview of the posterior base of leg II showing the distal region of muscle iiM7 (false-coloured in yellow) and its intermediate (white arrow) and distal (black arrow) attachment sites. The distal attachment site is shared with iiM14, iiM8, and iiM12 (not sectioned here). (b–d) Series of sections through the area around the intermediate attachment site (white arrows) of L7. White arrowheads indicate the muscle-muscle attachment between L6 and L7. Black arrowheads indicate the beginning of the proximal portion of muscle L7. Abbreviations: cu, cuticle; ep, epidermis; in, cuticular invagination; mi, mitochondria; ri, cuticular ridge. Scale bars: 1 μm (a); 500 nm (b–d).

Figure 11. Fine structure of iM14, with details of its proximal attachment site. STEM-in-SEM images. Sagittal sections; anterior is left, dorsal is up. (a) Overview of the posterior region of leg I showing iM14 and the distal region of iM7. (b) Close-up of the rectangular region in (a) showing the iM14 proximal attachment site (white arrow). A neuromuscular junction is adjacent to the attachment site. (c) Close-up of the rectangular region in (b) showing the neuromuscular junction, which is full of synaptic vesicles. Abbreviations: cg, claw gland; cu, cuticle; ep, epidermis; mi, mitochondria; ne, neuromuscular junction; nu nucleus. Scale bars: 1 μm (a); 500 nm (b); 200 nm (c).

Table 1. Nomenclature and serial homology of individual leg muscles in *Hypsibius exemplaris*.

Leg I	Leg II	Leg III	Leg IV
iM1	–	–	–
iM2	iiM2	iiiM2	ivM2
iM3	iiM3	iiiM3	–
iM4	iiM4	iiiM4	ivM4
iM5	iiM5	iiiM5	ivM5
iM6	iiM6	iiiM6	ivM6
iM7	iiM7	–	–
iM8	iiM8	iiiM8	ivM8
iM9	iiM9	iiiM9	ivM9
iM10	–	–	–
iM11	iiM11	iiiM11	ivM11
iM12	iiM12	iiiM12	ivM12
iM13	iiM13	iiiM13	ivM13
iM14	iiM14	iiiM14	ivM14
14	12	11	10

Supplementary Figure 1. Western blot showing the specificity of the DSHB 56-396-5 anti-myosin II monoclonal antibody using a lysate from pooled whole tardigrades. The tardigrade myosin II protein recognised by the antibody is ~250 kDa. The original, blotted gel (right lane) was stained with Brilliant Blue G 250 in order to judge blotting efficiency. (PDF)

Supplementary Movie 1. Video showing rotating leg muscles and an exploded view of leg musculature. 3D reconstructions based on CLSM data of F-actin labelling. (MP4)

Supplementary Movie 2. Video showing a cutaway distal view of the muscles of each leg. 3D reconstructions based on CLSM data of F-actin labelling. (MP4)

Supplementary Movie 3. Aligned series of STEM-in-SEM images showing the proximal attachment site of iM14. (AVI)

Supplementary File 1. CLSM stacks used for this study together with segmentation labels that were used to generate the 3D reconstructions. The resolution of stacks and their labels was reduced by half for the sake of file size. (ZIP file containing 3D-TIF files)

Supplementary Table 1. Comparison of leg muscle terminology from the literature and this study for *Hypsibius exemplaris* and closely related species. Note that labels from Müller [10] are compiled from several species. Labels are taken directly from references (from top to bottom): Müller [10] (*H. convergens*, *Ramazzottius oberhaueseri*, *Macrobotus hufelandi*), Schmidt-Rhaesa & Kulesa [11] (*Hypsibius* sp.), Smith & Jockusch [15] (*H. exemplaris*), Marchioro *et al.* [14] (*Acutuncus antarcticus*), this study (*H. exemplaris*). Question marks indicate ambiguous or missing data. See main manuscript for references. (PDF)

1
2
3
4
5
6
7
8
9
10
11
12
13
14
15
16
17
18
19
20
21
22
23
24
25
26
27
28

Appendix B

Please find our point-by-point responses to the reviewers **in bold** below.

Reviewer: 1

Summary: In this manuscript, the authors perform a very detailed and rigorous analysis of the leg musculature of the eutardigrade *Hypsibius exemplaris*. The analysis focuses on a very important problem that has not been adequately addressed in previous studies—identifying serially homologous muscles between the segmentally reiterated legs of a tardigrade. This study is all the more important, given that *H. exemplaris* has recently become the developmental model for evolutionary studies of tardigrades. This study provides a much-needed guide for recognizing particular segmental identities, the developmental basis of which may be unraveled in future developmental studies. Additionally, this study clearly demonstrates that intrinsic muscles are found in the legs of tardigrades, which may lead to a fairly large shift in our understanding of the transition of lobopodal legs to the jointed appendages characteristic of arthropods. The data and methods used to collect it are of very high quality, and the text is well written. This manuscript will be broadly interesting to biologists from several fields that are interested in the evolution or functional morphology of animal legs. I have a few comments that I hope that the authors find useful. Page numbers referenced below are in the very top left or right corners of the proof. Line numbers referenced below are on the left side of the proof.

Major comments:

1.) Section 4.2, page 6 of 22, lines ~34–36: In reference to the proximal attachment sites of M9 and M11, it is unclear what criterion is being used to delineate segment boundaries. I recommend that the authors explain how they determined that the proximal attachment sites for these muscles were in an anteriorly adjacent segment, and not in the anterior part of the same segment.

Although the exact segmental boundaries have not yet been completely clarified in tardigrades, we have made several changes in order to resolve the issue raised by the reviewer. In short, we use the segmental scheme illustrated in Smith and Jockusch (2014).

Specifically, we have corrected the description of the layout of the ventral attachment sites in our introduction (3rd paragraph) according to the reviewer's comment below (first point under "minor comments"). In this passage, we cite the article by Smith and Jockusch (2014) that illustrates all ventral attachment sites relative to the currently accepted segmental boundaries in tardigrades, and this is the same scheme we use in our manuscript.

Second, we have modified the passage the reviewer is referring to by specifying exactly which ventral attachment site each of these muscles attaches to (section 4.2, 2nd paragraph):

"In legs II–IV, each M9 muscle attaches proximally at the anterior ventral attachment site of its respective segment, while M11 attaches at the posterior ventral attachment site of this segment (figures 3 and 4*b–d*). In other words, iiM9, iiiM9, and ivM9 attach at ventral attachment sites 2, 4, and 6, respectively, while iiM11, iiiM11, and ivM11 attach at ventral attachment sites 3, 5, and 7, respectively (figures 2 and 3)"

In addition, we have labeled the ventral attachment sites in our figure 1.

2.) Figures 6 and 7, and associated text. Statements in text depend on accurately delineating muscle strands M6 and M7 as separate cells. This is done by counting nuclei—with 1 nuclei expected for each muscle cell. However, it is unclear how the authors determined which nuclei belong to which muscle strand (s). In the images they have arrowheads pointing to particular nuclei, but several other nuclei are also found in close proximity to the muscle fibers in question. I recommend that the authors explain how they identified the nuclei of particular muscles strands.

We have added the following statement to our results (section 4.1): "The nucleus belonging to each muscle is identifiable via myosin labelling, which stains both the contractile strand and the cell body surrounding the nucleus (figures 6a–d and 7a–d)."

Minor comments:

Section 2, page 3 of 22, line 48: The anteriormost ventral attachment site in *H. exemplaris* sits immediately behind the first trunk ganglion, so it is not in the head, as stated. I recommend modifying this statement.

We have corrected this statement to read as follows: "...two sites each in the segments bearing legs II–IV and one in the segment bearing leg I."

Section 4.1, page 6 of 22, line ~8: Refers to figure 1a–d, but figure 1 only includes panels a and b. Please fix this.

Done.

Section 4.2, page 6 of 22, line ~44: Figure 8a–d is referred to as evidence of a large internal haemolymph in the legs, but this figure primary shows SEMs of the external cuticle. I recommend deleting reference to this figure in this regard.

Done.

Section 4.2, page 6 of 22, line ~54: Referring to "(figure 4a, e). This reference seems to be to support the statement that iM11 attaches distally. However, iM11 is not labeled in figure 4a. I recommend labeling it, or rewriting the sentence in question to make it more clear what the reference to "(figure 4a, e)" is meant to support.

We have added the label for iM11 to figure 4.

Section 4.2, page 6 of 22, line ~56–bottom of page: Referring to lateral position within the legs of muscles M2, M3, and M4. Whether this attachment site is at the lateral part of the leg is difficult to determine based on the figures in text. The best demonstration of this statement is found in the inset model found in figure 8a. I recommend referencing this figure as evidence for the lateral location of the distal attachment sites of these muscles. Also, I recommend referencing any of the supplemental movies that show the distal location of these attachment sites.

We have added references to figure 8a and Supplementary Movie 1 for this passage.

Section 4.2, page 6 of 22, last paragraph: Referring to muscles M3 and M4. The text states that the distal attachment sites of these muscles are at “approximately half the length of the leg”, but they appear to attach in the proximal most part of the leg, much more proximal than stated. I recommend modifying the text to make this clear.

We have corrected this statement to read, "...at the base of the leg..."

Section 4.2, page 7 of 22, ~lines 3–4: Referring to the statement that the “nucleus of each M3 muscle is located between the two strands..”, with the associated reference to figure 6b. Figure 6b shows that the nuclei of both M2 and M5 are located between the split in these strands, but M3 is not split, and is not labeled in this panel. I suspect that the authors meant to refer to muscles M2 and M5, not M3. If so, I recommend making this change in text. If not, please modify the figure panel to show where muscle strand M3 is located.

We actually meant to refer just to muscle M5 in this section. We have corrected this in the text.

Section 4.2, page 7 of 22, ~lines 11–13: Referring to “(figure 7a–d)” as evidence that the “posteriormost strand of iiiM6 corresponds in attachment sites and course to iM7 and iiM7...”. I recommend also referencing “(Figure 3, 4a–c)”.

We have added the references to figures 3 and 4a–c in the text.

Section 4.2, page 7 of 22, ~line 19–20: Referring to shared attachment sites between M7 and M6. The figure reference should be “(figure 10a–c)”, not “(figure 10a–d)”, since M6 is not shown in panel d. I recommend making this change.

It is true that muscle M6 is mostly not visible in figure 10d, but a small part of the attachment structure between M6 and M7 is still present in this figure, indicated by the white arrow. Therefore, we prefer to keep the reference to the entire figure 10a–d in the text.

Reviewer: 2

General comment:

Aiming at understanding if tardigrade leg muscles are serially repeated, the authors investigated the overall arrangement as well as ultrastructural aspects of the leg musculature of *Hypsibius exemplaris*. In order to achieve their goals, authors used an approach based on (immuno)cytochemistry tools and advanced microscopy techniques.

All confocal laser scanning and electron micrographs and 3D reconstructions presented in the figure plates are of high quality, which certainly denotes the great effort made by the authors to perform this investigation. Great job!

The introduction to the questions about segmental patterns in Tardigrada, especially in what concerns leg myoanatomy, is well exposed and the methodology used is well described. However, the description of the findings is in general acceptable but needs to be revamped. Therefore, I recommend the authors to revise the ms. (with an extra care in Results) if its overall quality is to be increased. I certainly look forward to see this ms. published and the novel findings it encloses available to the scientific community.

General comments for the sections:

-Abstract: succinct and the questions that drove the authors are clearly defined. However, it lacks information about the conclusions. Please consider including the main results and conclusions from your study.

- Introduction + Material and Methods: well presented; I made some corrections/comments/suggestions directly on the submission PDF.

- Results: as compared to the rest of the ms., this section clearly lacks quality mainly due to an incomplete, vague description of the findings. As pointed out in several comments embedded in the PDF, it is important to provide the reader with a clear description of the arrangement of all muscles mentioned.

- Discussion: I find subsection 5.1 somewhat speculative in general and made a couple comments about it.

- Conclusions: well presented; it wraps up the whole story coherently.

Major & Minor remarks:

Several comments, corrections and suggestions have been annotated directly (and going straight to the point) on the submitted PDF. These remarks concern the text (including figure captions) and figures.

Note from the authors: Reviewer #2 made several comments/questions directly in the PDF of the submitted manuscript. These comments/questions are copied below directly from the PDF and addressed accordingly.

Section 1, page 3 of 22, line 20: You studied solely one species and this should be stated here.

We have added "... in the eutardigrade *Hypsibius exemplaris*" to this sentence.

Section 3, page 4, line 23: Please, provide a succinct description on how the tardigrades were maintained.

We have added the following sentence: "Briefly, tardigrades were kept in plastic Petri dishes in mineral water (Volvic, Danone Waters Deutschland GmbH, Frankfurt am Main, Germany) at 21°C and fed unicellular algae (*Chlorococcum* sp.)."

Section 3, page 4 of 22, line 25: Just for my curiosity, have you ever tried to anesthetize them in commercially available sparkling water? If so, have you used them for ICC experiments with good results?

Yes, sparkling water also works but from our experience, the tardigrades immediately shrivel up and only become distended after about 3–4 hours. Since the 60°C tap water method is much faster, we generally use this method. We have not noticed any difference in ICC results.

Section 3.1, page 4 of 22, line 28: Have you performed controls in which you incubate specimens with only the 2ary Ab, meaning to assess its specificity?

We have added the following sentence to the immunohistochemistry section of our results (3.1): " Negative controls, whereby the primary antibody is omitted from the staining protocol, result in an absence of staining." To test for the specificity of the anti-

myosin antibody, we performed Western blots as described in the text and shown in the supplementary materials.

Section 3.2, page 5 of 22, line 9: For the readers who are not familiar with this acronym, please provide the full name.

We have added the full name to the text.

Section 3.3, page 5 of 22, line 16: Consider replace the term by "anesthetised" as used above.

We have adjusted the text according to the reviewer's recommendation.

Section 3.4, page 5 of 22, line 54: What do you exactly mean with "further segmentation"? Is it segmentation of the digital 3D representations?

We used "further" simply to indicate an additional processing step. For the sake of clarity, we have modified this sentence to read, "...were subsequently chosen for segmentation...".

Section 4.1, page 6 of 22, line 7: Phalloidin's specificity is high, indeed, but can you be so sure that all muscles were stained?

We are confident that really all muscles are stained based on a total comparison of our methods (myosin labelling, light microscopy, etc.) and the fact that additional non-muscular, F-actin-containing structures were also stained, such as the Malpighian tubules and microvilli of the gut.

Section 4.1, page 6 of 22, line 26: You say it at the beginning of the sentence.

We have corrected this in the text.

Section 4.1, page 6 of 22, line 32: The first part of the sentence is awkward. Please rephrase.

We have rephrased this sentence.

Section 4.2, page 6 of 22, line 51: None of the mentioned muscles are labelled in figure 4a,e. Please correct this and provide the information on the starting point of these fibers.

We have changed the figure reference in this sentence to figure 3 instead of figure 4.

Section 4.2, page 6 of 22, line 51: But where are the start and end point of this muscle? Please, provide a complete description.

We have added the following sentence to clarify the attachment points of this muscle: "It attaches distally in the anterior tip of the leg and proximally along the dorsal longitudinal musculature."

Section 4.2, page 6 of 22, line 54: The iM11 is not labelled in figure 4a,e. Please correct.

We have added these labels to the figure.

Section 4.2, page 6 of 22, line 54: This description is confusing. This part must be re-written in order to provide full description for each of the legs, i.e., to consider the point where each of the M9 muscles emerges from and where it ends. For instance, iiM9 seemingly emerges from the ventral attachment site 2 and extends into leg 2 where it attaches to ventral attachment 3. In addition, it would be helpful to indicate in figure 2 the position of the ventral attachment points, which can be used as anatomical reference points.

We have modified and expanded the descriptions of muscles M9 and M11 considerably. It now reads as follows: "In legs II–IV, each M9 muscle attaches proximally at the anterior ventral attachment site of its respective segment, while M11 attaches at the posterior ventral attachment site of this segment (figures 3 and 4*b–d*). In other words, iiM9, iiiM9, and ivM9 attach at ventral attachment sites 2, 4, and 6, respectively, while iiM11, iiiM11, and ivM11 attach at ventral attachment sites 3, 5, and 7, respectively (figures 2 and 3). Distally, ii–ivM9 attach at the base and ii–ivM11 at the tip of each leg (figures 2 and 3)." In addition, we have added labelling to figure 2 to indicate the ventral attachment sites.

Section 4.2, page 6 of 22, line 55–56: Again, this doesn't say much about the arrangement of muscles M11 in each leg. It requires a better description (start and end points). Then you can stress the fact that in legs II-IV this muscle spans through the whole leg but does not extend into the adjacent anterior segment.

We have modified the description of muscles M11. Please see the comment above.

Section 4.2, page 6 of 22, line 58–59: Please explain better how the muscle bifurcates in relation to the D-V axis (e.g., the branching point is located...).

We have added the following to this sentence: "...branching at approximately half its length".

Section 4.2, page 6 of 22, line 60: But above you say that M2 muscles attach at the base of the legs. What is correct?

We have corrected this inconsistency. It now reads, "...at the base of the leg...".

Section 4.2, page 7 of 22, line 3: This is a vague description. Please, be more specific and describe where exactly the muscles attach. In this case it should be said, e.g., that i-iiiM5 muscles span dorso-ventrally and attach proximally to the dorsal longitudinal muscles located opposite to I-III legs, respectively.

We have added the following sentence in order to address the reviewer's concern: "These muscles are oriented almost perfectly along the dorsoventral axis in legs I–III, i.e., the proximal attachment points are located directly above the middle of each leg (figures 2, 3, and 4*a–c*)."

Section 4.2, page 7 of 22, line 8: See my comment above - it needs a more thorough description.

We have added the following statement to clarify the attachment sites of the M6 muscles: "In contrast to the M5 muscles, the M6 muscles are not as vertically oriented. Instead, the proximal attachment sites are always located further posteriorly than the distal attachment sites."

Section 4.2, page 7 of 22, line 9: But where exactly? It seems that is attached within the boundaries of the segment which leg IV belongs to and midway in the D-V axis.

We have added the following additional detail to the sentence describing the posteriormost strand of iiiM6: "...where it shares an attachment site with the lateral longitudinal musculature and muscle ivM4 (figures 2, 3, 4c, d, and 7c, d)."

Section 4.2, page 7 of 22, line 16: Again, be more specific and provide a better description on where these muscles attach proximally.

We have added "...of the posterior adjacent segment" to this description of the proximal attachment site.

Section 4.2, page 7 of 22, line 19: A better description of the attachments is required here. Note that the muscle-muscle attachment between iiM6 and iiM7 seems to be a double hemidesmosome. A two-hemidesmosome structure has been reported in *Batillipes* but for the myo-epidermal attachments (see Kristensen 1978, i.e., your reference [19]). It seems that also *Halobiotus crispae* is characterized by this condition, though the image is very small (Fig. 6C) in K.A. Halberg, D. Persson, N. Møbjerg, A. Wanninger, R.M. Kristensen Myoanatomy of the marine tardigrade *Halobiotus crispae* (Eutardigrada: Hypsibiidae) J. Morphol., 270 (2009) 996-1013. Interestingly, a very similar condition was found in Cycliophora (see figure 8B in Neves RC, Cunha MR, Kristensen RM, Wanninger A. 2010b. Comparative myoanatomy of cycliophoran life cycle stages. J Morphol 271: 596– 611.). To my knowledge, the term double hemidesmosome was used for the first time, and described, in "Listgarten MA. 1974. The double hemidesmosome: A new intercellular junction. Am J Anat 141:133–138." - see Figs 1 and 2 in this publication. Compare structures such as the lamina densa, lamina lucida and the cytoplasmic condensation shown in Listgarten's figures and compare with your Fig. 10b.

We thank the reviewer for calling our attention to these articles. We have added the following passage regarding the double hemidesmosome-like structures to our discussion (section 5.1): "The muscle attachments themselves superficially resemble the "double hemidesmosome" structures first described from rhesus monkey epithelial tissue by Listgarten [19, 40]. Interestingly, such structures have also been described from the cycliophoran *Symbion pandora* [41]. It is important to note that desmosomes and hemidesmosomes are typically associated with cytoplasmic intermediate filaments, which are, however, missing in panarthropods [26]. In *H. exemplaris*, a lamin-derived cytoplasmic intermediate filament (cytotardin) has been co-opted for this task and appears as a belt around the cell, but it occurs exclusively in ectodermal tissues such as the epidermis and foregut [26]. Which protein fills this role in mesodermal tissue is not entirely clear, but our and previous ultrastructural data (see figure 2 in Shaw [16]) suggest that the muscle attachments of tardigrades are associated with actin rather than intermediate filaments, at least on the side of the muscle cell. In this regard, tardigrade muscle attachments show a unique combination of features: an association with actin filaments like in a typical adherens junction, while a layer of dense material between

cells (presumably the basement membrane) resembles the extracellular space of a hemidesmosome (present study; [19])."

Section 4.2, page 7 of 22, line 21: It does not seem "distalmost" to me. Please check.

It is true that this muscle does not stretch all the way to the tip of the leg, probably partially due to the presence of the claw gland, but it is the distalmost muscle in the posterior region of the leg. Regardless, we have changed "distalmost" to "distal".

Section 4.2, page 7 of 22, line 22: Please, slow down the Movie 3. It would be helpful to follow the structures.

We have slowed this video from 10 FPS to 2 FPS.

Section 4.2, page 7 of 22, line 26: Please, provide a better description on the arrangement of M8 muscles, with a special attention to leg IV (which will make it easier to understand your point in subsection 4,3).

We now state that iiiM8 and ivM8 share a proximal attachment site.

Section 4.2, page 7 of 22, line 26: Also requires a better description for each leg.

We have now specified the ventral attachment sites for these muscles in each leg.

Section 4.2, page 7 of 22, line 27: What's the difference between medial and middle in this case? This is confusing. Please correct.

We have replaced "medial" with "closest to the midline of the body" in order to avoid confusion. We have also changed "lateral" to "lateralmost"

Section 4.2, page 7 of 22, line 49: I have no access to this paper, so there isn't much I can say about this comparison.

This manuscript was submitted as a companion paper and will be published in the same issue.

Section 4.2, page 7 of 22, line 54: Perhaps it would be better to stress that these muscles are V-shaped, contrary to the condition found in the ctenophores.

To our knowledge, there is no ultrastructural data of these muscles in ctenophores so while, yes, SEM images seem to show a branching pattern in the ctenophore that appears different from that in the nematomorph or tardigrade, we cannot say more about the implications or nature of this sort of structure. Our goal here was simply to point out that similarly derived morphologies of muscle cells have also been described from other animals.

Section 5.1, page 8 of 22, line 4: Do you mean "At the posterior region of each leg"?

We have modified this sentence accordingly.

Section 5.1, page 8 of 22, line 7: Aren't they too far from the edge of the leg to do so?

From our observations, the claws of eutardigrades fold in when the entire leg is retracted, probably passively due to contraction of the muscles that attach at the tip of the leg. When the entire M14 muscle contracts, the claws probably also fold in, at least partially.

Section 5.1, page 8 of 22, line 13: This is, at the minimum, highly questionable and speculative. Watching your Movie 1 makes me question such an assumption. Consider rephrasing to tone down.

We cannot find anything in Supplementary Movie 1 that contradicts our hypothesis. Regardless, we have added a statement explaining our reasoning, which reads as follows: " ...as it shows the same relative path and position of attachment sites as the M7 muscles"

Section 5.1, page 8 of 22, line 24: Why not "the posterior region of the legs"?

What we mean here is that there is a reduction in the total number of leg muscles between consecutive legs along the entire anterior-posterior body axis (i.e., number of leg muscles decreases from leg I to leg II to leg III to leg IV).

Section 5.1, page 8 of 22, line 54: Indicate the figure(s) in [10] where this muscle is depicted (it seems to be Figs. 5 and 7)

We have added a reference to this figure.

Section 5.1, page 8 of 22, line 58: Which strands? Please be more specific.

We are not referring to any specific muscle here but rather in general to the branched morphology that some muscles exhibit.

Figure 2 caption, page 11 of 22, line 8: Of what, F-actin labelling?

We have added "of F-actin labelling" to the figure caption.

Figure 4 caption, page 11 of 22, line 15: Please label muscles iM9, iM10, iM11.

We have added these labels as suggested.

Figure 4 caption, page 11 of 22, line 15: Do you mean Fig 2?

We have removed this sentence because we have now labelled all muscles also in figure 4.

Figure 4 caption, page 11 of 22, line 19: Please, label all muscles. The color code alone is not enough to easily locate each muscle.

We have added labels for a small number of muscles that are difficult to see in figure 4a–d, but we think labelling every muscle would instead add confusion to this set of

illustrations, which are simply intended to qualitatively show the similarity in the layout of muscles in each leg.

Figure 4 caption, page 11 of 22, line 22: You have two "orientations" in (e).

We have now specified that the left orientation is for legs I–III and the right orientation is for leg IV.

Figure 8 caption, page 11 of 22, line 43: Make a short note on the small insets (the one in the upper-right corner and the one in the lower-left corner) to explain their relation with the other images shown in this panel.

We have added a note to the figure caption that reads: "Insets (in *a* and *b*) show 3D reconstructions based on F-actin labelling of the same perspectives with corresponding regions outlined."

Figure 8 caption, page 11 of 22, line 44: Please explain that regions of interest are outlined and shown as close-ups.

We have added this note to the figure caption.

Figure 8 caption, page 11 of 22, line 47: Explain that (c) and (d) are close-ups from the rectangular areas in (b) and delete the dotted lines.

We have adjusted the figure and figure caption according to the reviewer's suggestion.

Figure 9 caption, page 11 of 22, line 50: Make a short note on the small inset in the upper-right corner to explain its relation with the other images shown in this panel.

We have added a note to the figure caption that reads: "Inset shows 3D reconstruction based on F-actin labelling of the same perspective with corresponding regions outlined."

Figure 9 caption, page 11 of 22, line 52: Please explain close-ups (I have inserted a suggestion).

We have added an explanation of the close-ups.

Figure 10, page 11 of 22, line 55: Make a note about the area of interest outlined and shown as a close-up in (b).

We have added a note about the outline region.

Figure 11, page 12 of 22, line 3: Explain "white arrows" also here.

We have added an explanation of the white arrows.

Figure 11, page 12 of 22, line 5: The image (c) seems to be out of focus. Consider editing this image in order to sharp it (or sharp its edge) a little?

We had already sharpened the image a little bit but this magnification approaches the resolution limit of our microscope and we are therefore unable to further improve the image quality without introducing artifacts.

Supplementary Movie 3 caption, page 12 of 22, line 39: This video must be slowed down in order to make it easier for the reader to follow the structures of interest.

We have slowed this video from 10 FPS to 2 FPS.

Figure 6c: Move the label (c) to the upper-right corner in order to give a full view of the image.

We have adjusted this figure according to the reviewer's suggestion.

Figure 6d: As I pointed out in the previous comment, move the label (d) to the upper-right corner in order to give a full view of the image.

We have adjusted this figure according to the reviewer's suggestion.